# Protriever: End-to-End Differentiable Protein Homology Search for Fitness Prediction

**Ruben Weitzman** [1 2] **Peter Mørch Groth** [3 4] **Lood Van Niekerk** [5] **Aoi Otani** [2] **Yarin Gal** [1] **Debora S. Marks** [2] **Pascal Notin** [2]

## Abstract

Retrieving homologous protein sequences is essential for a broad range of protein modeling tasks such as fitness prediction, protein design, structure modeling, and protein-protein interactions. Traditional workflows have relied on a two-step process: first retrieving homologs via Multiple Sequence Alignments (MSA), then training models on one or more of these alignments. However, MSA-based retrieval is computationally expensive, struggles with highly divergent sequences or complex insertions & deletions patterns, and operates independently of the downstream modeling objective. We introduce Protriever, an end-to-end differentiable framework that learns to retrieve relevant homologs while simultaneously training for the target task. When applied to protein fitness prediction, Protriever achieves state-of-the-art performance compared to sequence-based models that rely on MSA-based homolog retrieval, while being two orders of magnitude faster through efficient vector search. Protriever is both architecture- and task-agnostic, and can flexibly adapt to different retrieval strategies and protein databases at inference time – offering a scalable alternative to alignment-centric approaches.

## 1. Introduction

Proteins have evolved over billions of years under strict constraints and evolutionary pressure, where the preservation of function shapes the landscape of permissible mutations (Göbel et al., 1994). Understanding these mutational landscapes is fundamental to both basic biological research and the engineering of novel proteins for applications in therapeutics, new materials, and sustainability. Homologous proteins – sequences sharing a common evolutionary origin – are particularly valuable for modeling because they often exhibit similar structural and functional properties that reveal fundamental constraints on sequence variation. Consequently, leveraging homology has become essential for a range of protein modeling tasks, from predicting the effects of mutations on disease risk (Frazer et al., 2021), to guiding protein design campaigns (Russ et al., 2020; Notin et al., 2024), or inferring tertiary structure from sequence (Jumper et al., 2021).

Traditional protein modeling workflows follow a two-stage process: first retrieving homologs via Multiple Sequence Alignments (MSAs), then training models on one or more of these alignments. This approach has yielded family-specific models that effectively capture evolutionary constraints for particular protein families (Krogh, 1998; Hopf et al., 2014; Frazer et al., 2021). However, this paradigm suffers from several fundamental limitations. First, MSA-based retrieval often misses distantly related sequences that could provide valuable evolutionary context but fall below alignment significance thresholds. Second, sequences containing large insertions, deletions, or structural rearrangements remain difficult to align reliably, despite potential functional relatedness (Riley et al., 2023). Third, retrieval operates independently of downstream modeling objectives, relying on alignment heuristics rather than principled, data-driven approaches to identify the most informative homologs for specific tasks. Additionally, analyzing new protein families requires constructing separate MSAs and training distinct models, making the approach computationally demanding and poorly suited for large-scale applications.

Recent advances in large-scale protein language models (pLMs) have yielded flexible, alignment-free approaches that leverage the vast diversity of known protein sequences (Elnaggar et al., 2021; Lin et al., 2023; Nijkamp et al., 2023). However, single-sequence models often underperform family-specific methods for variant effect prediction,

---

[1]Department of Computer Science, University of Oxford [2]Department of Systems Biology, Harvard Medical School [3]Department of Computer Science, University of Copenhagen [4]Enzyme research, Novonesis [5]Ginkgo Bioworks. Correspondence to: Ruben Weitzman <ruben.weitzman@cs.ox.ac.uk>, Debora Marks <debbie@hms.harvard.edu>, Yarin Gal <yarin@cs.ox.ac.uk>, Pascal Notin <pascal_notin@hms.harvard.edu>.

*Proceedings of the 42$^{nd}$ International Conference on Machine Learning*, Vancouver, Canada. PMLR 267, 2025. Copyright 2025 by the author(s).

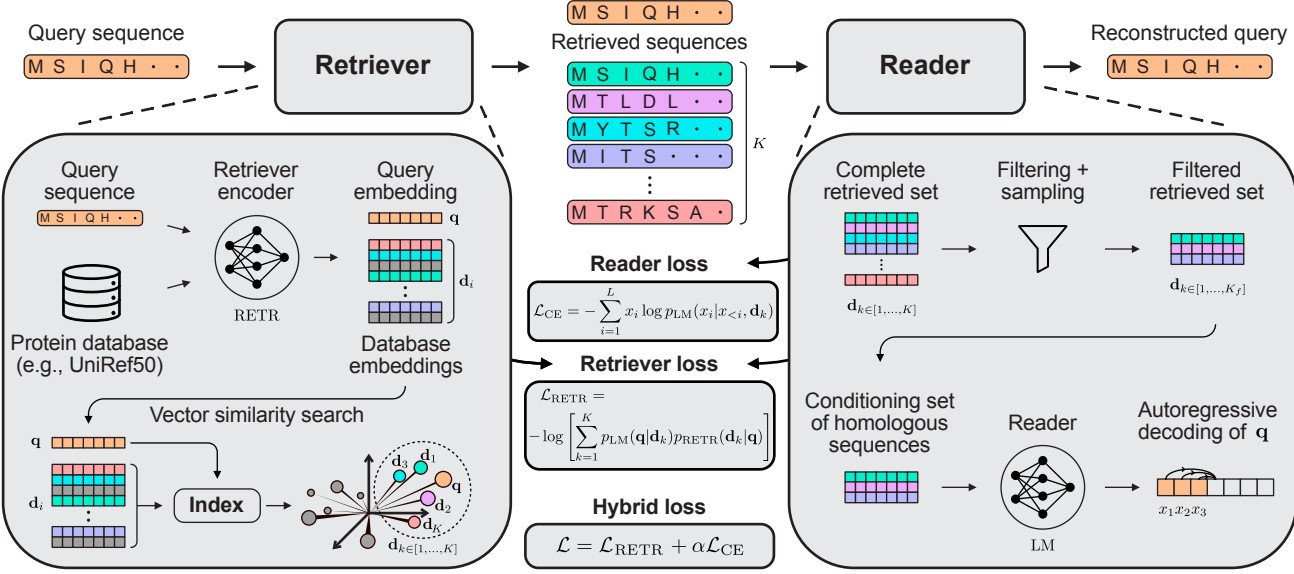

*Figure 1.* **Protriever**. The Protriever framework is composed of three parts: a learned retriever, an index, and a reader. The two neural networks work together to produce a conditional sequence likelihood. The retriever selects a set of sequences $\mathcal{D}_K = \{\mathbf{d}_k\}_{1,\dots,K}$ to be passed on to the reader using vector similarity search between the embedded query sequence and retrieval index. This set of sequences is then passed on to the reader which learns to reconstruct the query from the filtered conditioning set $\mathcal{D}_{K_f}$. During training, the reader calculates the relevance score of each document $p_{\text{LM}}(\mathbf{q} \mid \mathbf{d}_k)$ which are then used to train the retriever.

particularly when confronted with rare or highly specialized proteins (Notin et al., 2023). Hybrid solutions therefore integrate family-specific context through several approaches. Some methods fine-tune pretrained models on homologous sequences for specific protein families (Alley et al., 2019), while others learn distributions over fixed sets of retrieved sequences during training (Rao et al., 2021; Truong Jr & Bepler, 2023). A third strategy performs retrieval at inference time, integrating family-specific statistics with learned representations across families (Notin et al., 2022). Despite this progress, existing retrieval frameworks remain static, where retrieval depends on fixed notions of sequence similarity without allowing models to refine or backpropagate through retrieval choices.

In this work, we propose **Protriever**, an end-to-end differentiable protein homology search for fitness prediction. Our approach enables fast vector-based homology retrieval using learned dense representations and integrates these retrieved sequences to yield accurate zero-shot fitness predictions. Our contributions are as follows:

- We develop an end-to-end differentiable retrieval and sequence modeling approach, enabling our joint architecture to learn which homologous sequences are the most informative for the downstream task of interest (§ 3.1);

- We explore various hybrid losses for joint training, re-

triever pretraining strategies such as Dense Passage Retrieval (DPR), and different sampling strategies to optimize fitness prediction performance (§ 3.2 and § 4.1);

- We leverage and adapt several architectural speedups for fast vector search, including inverted file indexes and product quantization (§ 3.3);

- We demonstrate Protriever achieves state-of-the-art performance among sequence-based models on the ProteinGym benchmarks, while being orders of magnitude faster to retrieve homologs than standard MSA approaches including JackHMMER, MMseqs2, and MMseqs2-GPU (§ 4 and § 5).

The integration of retrieval mechanisms directly into the training process addresses crucial gaps in traditional protein sequence modeling. First, it overcomes the fixed nature of homology sets and their limitations on adapting to newly discovered or less common sequences. Second, by operating in dense representation space rather than requiring explicit alignments, Protriever can leverage distantly related or structurally divergent sequences that are difficult to align reliably but may still provide valuable evolutionary context. This dual advantage allows the model to dynamically discover informative homologs while accessing a broader range of evolutionary relationships than alignment-based approaches.

## 2. Related work

### 2.1. Alignment-based models

Alignment-based models have been the cornerstone of protein sequence analysis for decades. The traditional workflow involves searching large protein databases for homologs, constructing Multiple Sequence Alignments (MSAs), and fitting statistical models to capture evolutionary constraints within protein families. Early methods like PSSM (Position-Specific Scoring Matrices) and HMMs (Hidden Markov Models) extracted position-specific information from alignments (Krogh, 1998). Subsequent approaches captured pairwise co-evolutionary signals through energy-based models (Hopf et al., 2017), while more recent methods leverage variational autoencoders to model higher-order correlations between sequence positions (Riesselman et al., 2018; Frazer et al., 2021). However, alignment-based methods face fundamental limitations that have persisted across generations of approaches. MSA construction algorithms like MUS-CLE (Edgar, 2004), ClustalW (Thompson et al., 1994), and more recent tools like JackHMMER (Johnson et al., 2010) and MMseqs2 (Steinegger & Söding, 2017) rely on sequence similarity heuristics that can fail for highly divergent sequences. These methods struggle with regions containing extensive insertions, deletions, or structural rearrangements, often producing gaps that obscure true evolutionary relationships. Additionally, MSA-based models must be trained separately for each protein family, making them computationally expensive and limiting their applicability to families with sparse sequence data. The fixed nature of these alignments also means that as protein databases grow with newly sequenced genomes, existing models become obsolete and require complete MSA reconstruction and retraining.

### 2.2. Protein language models

Protein language models (pLMs) emerged as alignment-free alternatives, trained with self-supervised objectives on massive protein sequence databases. These models learn evolutionary constraints that generalize across families, particularly benefiting small families with limited homologs. UniRep (Alley et al., 2019) pioneered cross-family protein modeling using LSTM architectures. The field then adopted Transformer-based approaches, including autoregressive models like ProGen (Madani et al., 2020), Tranception (Notin et al., 2022), and ProtGPT2 (Ferruz et al., 2022), and masked language models like ESM (Rives et al., 2021), PRoBERTa (Nambiar et al., 2020), and ProtTrans (El-naggar et al., 2021). Despite their flexibility, pLMs often underperform family-specific models for fitness prediction and require substantial computational resources to encode evolutionary knowledge across all protein families in their parameters.

### 2.3. Hybrid fitness prediction models

Hybrid approaches combine the generalization capabilities of pLMs with evolutionary specificity through three main strategies. The first involves fine-tuning pretrained models with the same self-supervised objective on a subset of family-specific sequences, typically retrieved with a MSA, as demonstrated by UniRep's evolutionary fine-tuning (evotuning) (Alley et al., 2019) or ESM-1v's spiked fine-tuning (Meier et al., 2021). The second strategy trains models across sets of homologous sequences during training. MSA Transformer (Rao et al., 2021) learns distributions over entire MSAs using axial attention and masked language modeling, enabling rich cross-family representations. PoET (Truong Jr & Bepler, 2023) and ProtMamba (Sgarbossa et al., 2024) eliminate alignment requirements by learning distributions over concatenated homologous sequences. The third approach performs retrieval at inference time. Tranception (Notin et al., 2022) weights predictions from an unconditional language model with position-specific frequencies from MSAs, while TranceptEVE further integrates predictions based on a model that learns dependencies across positions in the MSA. However, all these approaches rely on predefined homology sets determined by traditional sequence similarity heuristics, failing to leverage model-learned representations for optimal retrieval.

### 2.4. Retrieval

Retrieval methods identify relevant database objects to improve task performance. In NLP, retrieval has enhanced open-domain question answering (Robertson & Zaragoza, 2009; Grave et al., 2017; Wang et al., 2018; Karpukhin et al., 2020) and knowledge-grounded text generation (Lee et al., 2019). Advanced methods like REALM (Guu et al., 2020) and RAG (Lewis et al., 2020) demonstrated end-to-end differentiable training of retriever-reader architectures for retrieval-augmented generation. In protein modeling, MSA generation has been framed as retrieval, with some work exploring differentiable MSA construction (Hong et al., 2021; Petti et al., 2023; Llinares-López et al., 2023). Recent approaches include AIDO.RAG (Li et al., 2024), which trains retrievers for improved MSA generation by using hierarchical clustering to generate sequence identifiers that a causal language model learns to predict, and RSA (Ma et al., 2024), which uses frozen ESM-1b encoders for sequence retrieval. However, no prior work has achieved end-to-end joint training of retrieval and protein sequence modeling.

## 3. Protriever

### 3.1. Overall architecture

The Protriever framework is comprised of three components: the Retriever model, the Index, and the Reader model (see

Figure 1). At a high level, the query sequence is first processed by the retriever, which performs similarity search against a fixed index of sequence embeddings to identify relevant homologs. The reader model then performs the target task conditioning on the retrieved homologous sequences. During training, the target task is typically a self-supervised objective such as autoregressive decoding of the query sequence. The reader learns which retrieved sequences provide useful context for that task, providing gradient feedback to the retriever that adjusts sequence relationships in embedding space accordingly. The following subsections detail each component and the training procedure.

## 3.2. Retriever module

**Initialization** We use a transformer encoder architecture for the retriever, which we initialize with ESM-2 (Lin et al., 2023) pre-trained weights (35M parameters). Average pooling is applied over the outputs of the last layer to obtain a 480-dimensional vector representation for each sequence. To then compute similarity between a query $\mathbf{q}$ and other sequences in the index, we compute the cosine similarity $s(\mathbf{d}, \mathbf{q})$ between their corresponding embeddings.

**Pretraining with dense passage retrieval** Following the dense passage retrieval paradigm (DPR, Karpukhin et al. (2020)), we further pretrain the retriever encoder to learn effective protein sequence representations in a low-dimensional continuous space. The core objective is to create an embedding space where homologous protein sequences have higher similarity scores than non-homologous ones. We construct training data from UniRef50 using BLAST all-vs-all search to identify homologous sequences for each query (see Appendix G). Let $\mathcal{D} = \left\{ \langle \mathbf{q}_i, \mathbf{d}_{i,1}^+, \ldots, \mathbf{d}_{i,m_i}^+, \mathbf{d}_{i,1}^-, \ldots, \mathbf{d}_{i,n}^- \rangle \right\}_{i=1}^M$ be the training data consisting of $M$ instances. Each instance contains one query sequence $\mathbf{q}_i$, $m_i$ relevant (positive) homologous sequences $\mathbf{d}_{i,j}^+$, and $n$ irrelevant (negative) sequences $\mathbf{d}_{i,j,k}^-$. Given this data, we optimize the encoder parameters to maximize the similarity $s(\mathbf{d}^+, \mathbf{q})$ while minimizing $s(\mathbf{d}^-, \mathbf{q})$ using the negative log likelihood:

$$\mathcal{L}_{\text{pretrain}} = -\sum_{i=1}^M \sum_{j=1}^{m_i} \log \frac{e^{s\left(\mathbf{q}_i, \mathbf{d}_{i,j}^+\right)}}{e^{s\left(\mathbf{q}_i, \mathbf{d}_{i,j}^+\right)} + \sum_{k=1}^n e^{s\left(\mathbf{q}_i, \mathbf{d}_{i,k}^-\right)}} \tag{1}$$

Negative examples are comprised of both random sequences and positive examples for other queries from the same mini-batch. During training, we sample UniRef50 clusters with weight inversely proportional to cluster size to avoid over-representing large clusters (see Figure G.1). We then replace each UniRef50 sequence, whether query or cluster member, with a UniRef100 sequence with weight inversely propor-

tional to the size of the corresponding UniRef90 cluster. As data augmentation, we randomly reverse query sequences during training from either N-terminus to C-terminus, or vice versa, to make retrieval agnostic to sequence order.

This pretraining phase establishes a foundation for the subsequent end-to-end training of the full Protriever architecture, ensuring that the retriever can effectively identify relevant protein sequences from the UniRef50 index for downstream reconstruction tasks.

## 3.3. Index

The index consists of embeddings for all UniRef50 (Suzek et al., 2015) sequences ($\approx 62$ million), computed using the (pretrained) retriever encoder and stored for efficient similarity search via Faiss (Johnson et al., 2021). As the retriever encoder is updated during training, the index embeddings can become stale relative to the evolving retriever representations. Since updating the index is computationally expensive—requiring re-encoding all 62 million sequences—we update it 10 times throughout training (every 5k training steps). This approach balances maintaining reasonably current embeddings with computational efficiency.

For efficient search at scale, we implement two key optimizations. First, we use an inverted file index (IVF), where the embedding database is partitioned using $k$-means clustering into $K_{\text{IVF}}$ clusters. At query time, we search only the nearest $P_{\text{IVF}}$ clusters rather than the entire database. Second, we apply product quantization (PQ) to compress the stored vectors, reducing memory usage while maintaining retrieval accuracy. Due to the scale of UniRef50, the index is distributed across multiple GPUs, with queries processed independently on each partition and results aggregated to produce the final retrieval set. Detailed specifications are provided in Appendix A.

## 3.4. Reader module

We use PoET (Protein Evolutionary Transformer) (Truong Jr & Bepler, 2023) as our reader backbone. PoET is designed to operate on sets of homologous sequences by processing them as concatenated sequences-of-sequences through a specialized decoder transformer architecture. Unlike alignment-based approaches, PoET learns to model evolutionary relationships without requiring explicit sequence alignment, making it well-suited for integration with our retrieval framework. The PoET architecture processes the query sequence concatenated with all retrieved homologous sequences simultaneously, allowing the model to capture complex interactions between all positions across the entire set. This enables the reader to learn rich representations that incorporate evolutionary context from the retrieved sequences when making predictions for the query.

We initialize our reader with a pretrained PoET model that was trained on the same UniRef50 dataset used for our retriever's DPR pretraining. This initialization provides a strong foundation for subsequent end-to-end training, as the model already understands how to process sets of homologous sequences effectively. Our framework is general and can adapt to various model architectures and self-supervised objectives, as long as these architectures can condition their predictions on sets of homologous sequences. We present results with an efficient sequence-to-sequence architecture (Fusion-in-Decoder) in Appendix B, demonstrating the versatility of our approach.

### 3.5. Protriever training

We evaluate different loss functions for end-to-end training of the retriever. Our approach leverages the language model's performance to guide retriever training: if a homologous protein proves valuable for the reader's sequence modeling task, the retriever is encouraged to rank it closer to the query sequence in embedding space.

The relevance score of a protein sequence $\mathbf{d}$ to a query sequence $\mathbf{q}$ is computed as:

$$p_{\text{RETR}}(\mathbf{d} \mid \mathbf{q}) = \frac{\exp(s(\mathbf{d}, \mathbf{q})/\theta)}{\sum_{k=1}^{K} \exp\left(s\left(\mathbf{d}_k, \mathbf{q}\right)/\theta\right)} \quad (2)$$

where the sum is over $\mathcal{D}_K = \{\mathbf{d}_k\}_{1,\ldots,K}$ top-$K$ retrieved sequences, approximating the true relevance score over the entire index while maintaining computational tractability.

For our main experiments, we use the End-to-end training of Multi-Document Reader and Retriever (EMDR) loss (Sachan et al., 2021), which treats retrieved sequences as latent variables. Given a query $\mathbf{q}$ and the set $\mathcal{D}_K$ of top-$K$ retrieved sequences, the loss is:

$$\mathcal{L}_{\text{EMDR}} = -\log \left[ \sum_{k=1}^{K} p_{\text{LM}}\left(\mathbf{q} \mid \mathbf{d}_k\right) p_{\text{RETR}}\left(\mathbf{d}_k \mid \mathbf{q}\right) \right] \tag{3}$$

During optimization, we apply a stop-gradient operator to $p_{\text{LM}}$, ensuring updates are limited to the retriever parameters. In earlier stages of model development, we also experimented with alternative loss functions, including Perplexity Distillation (PDist) and Leave-One-Out Perplexity Distillation (LOOP) (see Appendix D).

## 4. Fitness prediction with Protriever

### 4.1. Fitness scoring methodology

The reader model of Protriever is trained with a conditional autoregressive objective, predicting the next token in a sequence of amino acids given retrieved sequences as context. The standard autoregressive factorization is extended by conditioning on a set of $K$ retrieved sequences $\mathcal{D}_K = \text{top-}K(P_{\text{RETR}}(\mathbf{d}|x))$:

$$P(x) = P_{\text{RETR}}(\mathcal{D}_K|x) \prod_{i=1}^{l} P_{\text{LM}}(x_i|x_{<i}, \mathcal{D}_K).$$

Following Frazer et al. (2021) and Notin et al. (2022), we evaluate the fitness of a mutated protein sequence $x^{\text{mut}}$ via its log-likelihood ratio with respect to the wild-type sequence $x^{\text{wt}}$:

$$F_x = \log \frac{P\left(x^{\text{mut}}\right)}{P\left(x^{\text{wt}}\right)}.$$

In practice, if both $x^{\text{mut}}$ and $x^{\text{wt}}$ are close in sequence space (e.g., differ by a handful of mutated positions), they will share the same conditioning set $\mathcal{D}_K$. In that case, the retriever probabilities cancel out, simplifying the fitness score to:

$$F_x = \log \frac{P_{\text{LM}}\left(x^{\text{mut}}|\mathcal{D}_K\right)}{P_{\text{LM}}\left(x^{\text{wt}}|\mathcal{D}_K\right)}.$$

To apply this scoring methodology, we first build an index of all protein sequences in our database. At inference time, we use the trained retriever from Protriever to encode all 62 million UniRef50 sequences. This process is parallelized across GPUs and uses FlashAttention (Dao et al., 2022) to enable large batch sizes, completing in approximately 30 minutes on four A100 GPUs. We then construct a Faiss index for fast similarity search in 3-4 minutes across four GPUs (see details in Appendix A).

Given a query sequence, we retrieve relevant homologs through a procedure inspired by re-ranking approaches in RAG (Izacard et al., 2022) and diversity maximization strategies in alignment-based protein modeling (Hopf et al., 2017; Rao et al., 2021). Re-ranking allows us to refine initial retrieval results by considering additional factors beyond similarity scores—in our case, diversity maximization to ensures we capture a broad range of evolutionary relationships. Following PoET (Truong Jr & Bepler, 2023), we enhance fitness prediction performance by ensembling $F_x$ estimates across multiple conditioning sets $\mathcal{D}_K$ that vary in size and composition.

More specifically, we search the UniRef50 index for homologous sequences and pull corresponding sequences from UniRef100. We sample based on effective cluster size and distance to query, evaluating five parameter combinations, along with three conditioning set sizes (6k, 12k, and 24k tokens), yielding an ensembling strategy across 15 forward

*Table 1.* **Zero-shot performance on the 217 substitution DMS of ProteinGym benchmark**. Reported metrics are Spearman rank correlation, AUC, MCC, top recall, and NDCG. Models are classified according to if they take as input MSAs (alignment based and Hybrid) or not (unconditional pLMs and Protriever)

| Model type | Model name | Spearman | AUC | MCC | Recall | NDCG |
|---|---|---|---|---|---|---|
| Alignment-based | Site independent | 0.359 | 0.696 | 0.287 | 0.201 | 0.748 |
| | GEMME | 0.459 | 0.749 | 0.353 | 0.211 | 0.777 |
| | EVE | 0.439 | 0.742 | 0.342 | **0.229** | 0.782 |
| Unconditional pLM | ESM-1v | 0.407 | 0.724 | 0.321 | 0.210 | 0.749 |
| | ProGen2 | 0.391 | 0.717 | 0.306 | 0.198 | 0.767 |
| | ESM2 | 0.405 | 0.726 | 0.322 | 0.213 | 0.764 |
| Hybrid | MSA Transformer | 0.432 | 0.737 | 0.341 | 0.223 | 0.777 |
| | Tranception L | 0.434 | 0.741 | 0.341 | 0.220 | 0.779 |
| | TranceptEVE L | 0.458 | 0.754 | 0.356 | **0.229** | 0.786 |
| | PoET | 0.470 | 0.759 | 0.368 | 0.226 | 0.784 |
| Protriever | Protriever | **0.479** | **0.762** | **0.374** | **0.229** | **0.788** |

passes with different configurations. Additional details on the sampling procedure are provided in Appendix E.

### 4.2. Experimental setup

We evaluate Protriever on the substitution benchmark of ProteinGym (Notin et al., 2023), containing 217 deep mutational scanning (DMS) experiments that probe the natural function of protein variants. DMS experiments systematically measure the functional effects of individual amino acid substitutions across a protein sequence, providing comprehensive fitness landscapes for specific proteins. Consequently, to perform well on this benchmark, models must capture a nuanced understanding of the biochemical constraints for the corresponding proteins as they must be able to detect subtle effects resulting from minor sequence changes.

Performance is evaluated using multiple metrics: Spearman correlation, AUC, and MCC capture broad performance across the full assay (important for general mutation effect prediction), while NDCG and top-K recall focus on the top end of the measured phenotypes (and thus most important for protein design applications). To evaluate the zero-shot capability of our retrieval framework, we score all sequences within each DMS using the same conditioning sets with homologous sequences as described above. Additionally, we score sequences in both directions (N-terminus to C-terminus and vice versa), a strategy shown to improve predictive performance (Notin et al., 2022).

### 4.3. Results

Protriever achieves state-of-the-art performance among sequence-based models on ProteinGym, as shown in Ta-

ble 1. Our approach reaches a Spearman correlation of 0.479 across all assays (aggregated by protein ID), outperforming the previous best sequence-only model PoET (0.470) and all other baselines. Protriever achieves the best performance across all metrics: AUC (0.762), MCC (0.374), NDCG (0.788), and top-K recall (0.229).

These results demonstrate that end-to-end differentiable retrieval can effectively identify and leverage homologous sequences for fitness prediction, achieving competitive performance with hybrid models that rely on traditional MSA construction while maintaining the computational efficiency and flexibility of learned retrieval. Additional results segmented by MSA depth and taxonomic groups are provided in Tables C.1 and C.2, showing that Protriever performs consistently well across different protein families and evolutionary contexts, with particularly strong performance on prokaryotes and viruses.

## 5. Discussion

### 5.1. Model training ablations

We conduct an ablation study to understand the individual contributions of DPR initialization and end-to-end joint training in our Protriever framework. Table 2 compares four experimental configurations in terms of Spearman correlation on the ProteinGym DMS substitution benchmark across different MSA depth regimes.

Using ESM embeddings directly as a retriever with our inference filtering and sampling scheme achieves a Spearman correlation of 0.432. This baseline demonstrates that ESM embeddings capture meaningful sequence homology relationships despite not being explicitly trained for retrieval

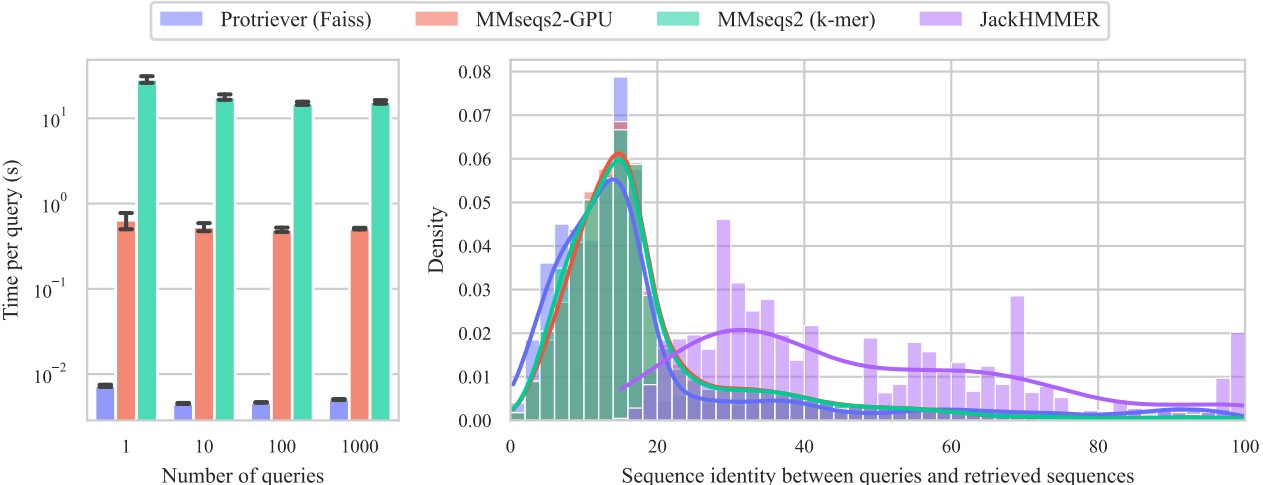

*Figure 2.* **Retrieval speed and quality**. *Left*: Retrieval time per query sequence (mean and standard error) at different query sizes using embedding similarity search, MMseqs2, and GPU-accelerated MMseqs2. Benchmarking details and tabulated values can be found in Appendix A.3 and Table A.2, respectively. *Right*: Distribution of sequence identities between queries and retrieved sequences for Protriever, MMSeqs2, and JackHMMER, using the ten ProteinGym validation sets as queries. The retrieved sequences have not been filtered and are weighed by number of retrieved sequences per method/query pair. Retrieval methods are sensitive to hyperparameters. We provide additional results for JackHMMER with bit scores 0.1, 0.5 (shown above), and 0.9 in Appendix A.4.

tasks, and that our retrieval framework can effectively leverage these relationships for fitness prediction.

Replacing the ESM retriever with DPR improves performance to 0.440, representing a substantial gain. This improvement highlights the advantage of contrastive learning for sequence retrieval, as DPR is specifically trained on known distant sequence homologs. Training the retriever and reader jointly (Protriever without DPR initialization) achieves 0.466, outperforming previous baselines. This demonstrates that joint optimization allows the retriever to learn representations better suited for the downstream fitness prediction task, with the reader model distilling information about homolog usefulness for sequence reconstruction across protein families.

Our full Protriever model, which combines DPR initialization with end-to-end joint training, achieves the highest performance at 0.479 Spearman correlation, validating our hypothesis that both components are essential. Analyzing performance by MSA depth reveals that both DPR initialization and joint training provide their largest benefits for low-depth MSAs, followed by medium-depth, then high-depth MSAs. This pattern suggests that learned retrieval is most valuable when traditional alignment-based methods struggle due to limited evolutionary information.

### 5.2. Inference-time speedups

A key advantage of Protriever is the substantial speedup gained by replacing MSA-based retrieval with vector sim-

ilarity search. The MSA sequences used by methods like EVE (Frazer et al., 2021) rely on sensitive retrieval tools such as JackHMMER (Johnson et al., 2010), which can take hours or days to complete for a given query sequence. Other retrieval routines such as BLAST (Altschul et al., 1990) or MMseqs2 (Steinegger & Söding, 2017) can offer speed improvements but at the cost of lower retrieval precision. Recently, MMseqs2-GPU (Kallenborn et al., 2025) further reduced search times to hundreds of milliseconds with marginal impact on retrieval accuracy. In contrast, our approach leverages fast vector similarity search using the Faiss library (Douze et al., 2024). Given the pre-trained index, retrieval is rapid, lightweight, and scalable as demonstrated in Appendix A.

To compare inference-time retrieval speed, we benchmark Protriever, MMseqs2 (Steinegger & Söding, 2017), and GPU-accelerated MMseqs2 (Kallenborn et al., 2025). The results are visualized in the left-hand side of Figure 2 and the benchmarking methodology is described in Appendix A.3. Our approach is two orders of magnitude faster than MMseqs2-GPU, enabling significant efficiency gains for large-scale applications such as proteome-wide predictions.

We benchmark four methods to compare inference time retrieval speed and downstream performance: Protriever, MMseqs2, GPU-accelerated MMseqs2, and JackHMMER (Potter et al., 2018). We search over UniRef50 clusters and apply a similar inference scheme as Protriever from Section 4.1. We apply the same 15% sequence identity cluster

*Table 2.* **Protriever model performance on substitution DMS benchmark**. Average Spearman's rank correlation between model scores and experimental measurements by MSA depth for different retrieval architectures.

| Experiment | End-to-End | DPR | Spearman by MSA depth | | | |
|---|---|---|---|---|---|---|
| | | | Low | Medium | High | Average |
| Frozen ESM | ✗ | ✗ | 0.368 | 0.439 | 0.485 | 0.432 |
| Frozen DPR | ✗ | ✓ | 0.403 | 0.452 | 0.484 | 0.440 |
| Protriever w/o DPR | ✓ | ✗ | 0.461 | 0.476 | 0.508 | 0.466 |
| Protriever | ✓ | ✓ | **0.464** | **0.498** | **0.512** | **0.479** |

filter as for Protriever, and sample according to combination of cluster sizes and distance to the query, where distance is given by sequence similarity instead of embedding distance (details in Appendix E). PoET serves as the reader model for all four configurations, with end-to-end training applied for Protriever.

Spearman's rank correlation and average per-query retrieval time for the ProteinGym substitution benchmark are shown in Table 3. The rapid retrieval of Protriever comes without performance loss for fitness prediction and even leads to gains through the joint retriever-reader framework. The CPU and GPU versions of MMseqs2 show nearly identical downstream performance as their hyperparameters were set to achieve similar sensitivity (see Appendix A.3 for details). JackHMMER, despite being a higher sensitivity search tool, achieves slightly lower Spearman correlation, consistent with observations in the PoET paper (Truong Jr & Bepler, 2023). This may reflect how traditional search methods make retrieval decisions independently of the downstream task and reader model architecture, potentially leading to suboptimal sequence selection. This further underscores the importance of joint training to align retrieval with downstream task modeling.

### 5.3. Qualitative analysis of retrieved sequences

While we have demonstrated significant speedup in inference time retrieval and improved downstream predictive performance, we have yet to examine the homology characteristics of the retrieved sequences. For ten validation sets (see Appendix F), we use wild-type sequences as queries and retrieve homologs using the same four methods as above. We compute sequence identity between queries and retrieved sequences and visualize the distributions on the right-hand side of Figure 2.

We observe that the distributions for MMseqs2 and Protriever searches have considerable overlap, while JackHMMER results show substantially higher similarities to the queries. However, JackHMMER's more sensitive results come at significant computational cost as shown in Table 3. It should also be noted that the four shown methods retrieve

different numbers of sequences with JackHMMER often returning comparatively few high-identity sequences, whereas MMseqs2 and Protriever retrieve larger numbers of more diverse sequences as shown in Figure 2, which partly explains JackHMMER's sequence identity distribution. The distributions after applying our standard 15% sequence identity filter (used during inference) are shown in Figure A.5. To further explore JackHMMER's sensitivity-performance trade-offs discussed above, we include results with varying sensitivity settings both before and after filtering in Appendix A.4.

Interestingly, despite JackHMMER retrieving sequences with higher evolutionary similarity, it does not achieve superior fitness prediction performance. This suggests that the relationship between sequence similarity and utility for fitness prediction is not straightforward, and that learned retrieval representations may capture different aspects of protein relationships that are valuable for downstream tasks.

### 5.4. Portability and Modularity

Our vector-based retrieval framework introduces a paradigm shift in how homology search is performed and deployed. Unlike traditional MSA-based methods that require computationally expensive sequence alignment at inference time, Protriever separates the indexing phase from the retrieval phase, enabling unprecedented portability and scalability.

The core advantage lies in the transferable vector index. Once protein sequences are encoded in embedding space, the resulting index is lightweight, portable, and can be easily distributed or shared across different computational environments. Our pre-computed index can be transferred as a compact file and immediately used for rapid retrieval without any additional preprocessing (see Appendix A, `IVFPQ96x8` indexing of UniRef50 uses 12.6GB of memory).

This architecture enables dynamic database management without retraining. New sequences can be encoded and added to the index incrementally, while outdated or erroneous entries can be removed instantly. This flexibility is particularly valuable for constantly evolving sequence databases driven by advances in sequencing technology,

*Table 3.* **Retrieval methods performance comparison**. Comparison of retrieval methods on ProteinGym substitution benchmark with retrieval timing. Inference is conducted using the scheme described in Section 4.1 with retrieval on UniRef50. Performance for MMseqs2 is therefore different than as reported in original paper or Table. The reader model for all four entries is PoET, which for Protriever is trained end-to-end using our framework.

| Retrieval method | Spearman by MSA depth (↑) | | | | Retrieval time (s) (↓) |
|---|---|---|---|---|---|
| | Low | Medium | High | Average | |
| Protriever | **0.464** | **0.498** | **0.512** | **0.479** | **0.0046** |
| MMseqs2 (k-mer) | 0.455 | 0.472 | 0.489 | 0.463 | 16.860 |
| MMseqs2-GPU | 0.454 | 0.470 | 0.491 | 0.462 | 0.613 |
| JackHMMER | 0.442 | 0.471 | 0.493 | 0.459 | 2501 |

or when working with proprietary sequences that cannot be publicly shared but can be incorporated into local indices. This approach contrasts with standard protein language models that encode sequence database information in their weights, making it difficult to leverage the information from newly added sequences without additional training.

The framework also supports domain-specific specialization. The same trained retriever can be applied to focused databases like GISAID (Shu & McCauley, 2017) for viral sequences or proprietary datasets for industrial applications, simply by re-indexing the target database without model retraining. This allows organizations to leverage the trained model while maintaining data privacy and domain relevance.

The framework is also model-agnostic. The retrieval framework can be integrated with various reader architectures, from encoder-decoder models (Appendix B) to decoder-only architectures. The retrieval encoder itself can be substituted with structure-aware models for tasks where structural information is crucial, where structural homologs for training could be generated with TM-Vec (Hamamsy et al., 2024).

This modularity extends to training objectives. While we focus on autoregressive sequence reconstruction, the end-to-end differentiable framework can be adapted for other self-supervised tasks such as masked language modeling on the query sequence, or supervised tasks such as property (e.g., thermostability, solubility) or tertiary structure predictions.

## 6. Conclusion

We have introduced Protriever, an end-to-end differentiable protein homology search framework that achieves state-of-the-art performance among sequence-based methods for protein fitness prediction while delivering computational speedups of two orders of magnitude over existing methods for homology search. The key innovation lies in joint training of retrieval and prediction components, allowing the retriever to learn which homologous sequences are most informative for the specific downstream task rather than relying on traditional task-independent retrieval routines.

This end-to-end optimization enables the model to discover functionally relevant evolutionary relationships that may be missed by alignment-based approaches. Our modular framework design allows the trained retriever to be paired with different reader architectures and adapted to various tasks beyond fitness prediction. This flexibility, combined with the computational efficiency of vector-based search, makes our approach broadly applicable across protein modeling applications. Additionally, our retrieval-based approach offers enhanced interpretability by allowing analysis of the sequences the model selects for conditioning, compared to traditional protein language models.

Future work will focus on scaling to larger databases and exploring how the sequences retrieved by our learned approach compare to those selected by traditional MSA methods, beyond sequence identity, providing deeper insights into the evolutionary relationships that drive protein function.

## Acknowledgements

We thank the members of the OATML group and Marks Lab for helpful discussions when writing this manuscript. R.W. is supported by the EPSRC Centre for Doctoral Training in Health Data Science (EP/S02428X/1). P.M.G. was supported by Innovation Fund Denmark (1044-00158A). Y.G. holds a Turing AI Fellowship (Phase 1) at the Alan Turing Institute, which is supported by EPSRC grant reference V030302/1. D.S.M. and P.N. were supported by a Chan Zuckerberg Initiative Award (Neurodegeneration Challenge Network, CZI2018-191853). This research was also supported by a Dean's Innovation Award for the Use of Artificial Intelligence in Research from Harvard Medical School.

## Impact statement

The advancement of protein design technologies promises transformative applications across many domains such as novel therapeutics, new materials, and sustainability. However, as these technologies evolve, they raise important dual-

use concerns regarding their potential applications. While protein fitness and design models can accelerate therapeutic discovery, the same underlying capabilities could potentially be misused for harmful purposes, including the design of biological weapons. A notable example of this risk is how a model trained to minimize protein toxicity for drug development could theoretically be modified to maximize toxicity instead (Urbina et al., 2022).

Our retrieval-based approach offers potential advantages for addressing some of these concerns. The dynamic nature of RAG systems means we can extend the conditioning database without retraining, separating the model from the knowledge base. This architecture is particularly valuable for proprietary or sensitive biological sequences—organizations can use the pretrained Protriever while maintaining their sequences in private, access-controlled databases rather than incorporating them into model weights during training. Such flexibility allows organizations to leverage the model's capabilities while maintaining strict control over their sequence data, which is especially important in biotechnology applications where data access may be restricted due to intellectual property concerns or biosafety considerations.

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

# Appendix

## A. Vector similarity search with Faiss

We rely on Faiss for GPU-accelerated vector similarity search (Johnson et al., 2021; Douze et al., 2024), whose terminology we adopt. The vector similarity search is facilitated with an *index*, whose task it is to search a large database of vectors, $\mathbf{d}$ and return the $K$ most similar ones to the query, $\mathbf{q}$, given a similarity metric.

The most simple index is a *flat* index, where the query is compared to all database entries. With the commonly used maximum inner product similarity measure, this reduces to computing $\mathbf{q} \cdot \mathbf{d}^T$ and extracting the $K$ largest entries. While this search is exact, it is both slow and requires storing all database vectors in memory which is prohibitively expensive. For fast search, an inverted file index (IVF) can be used. Prior to searching the index, all entries are clustered using a "coarse quantizer", e.g., a k-means clustering algorithm, given some predefined number of centroids, $K_{\text{IVF}}$. At search time, the query $\mathbf{q}$ is compared to all $K_{\text{IVF}}$ centroids, of which the $P_{\text{IVF}}$ most similar centroids, often referred to as the number of *probes*, are searched, reducing the number of comparisons from $N$ to

$$N_{\text{comparisons}} = K_{\text{IVF}} + P_{\text{IVF}} \frac{N}{K_{\text{IVF}}},$$

as per equation 18 in Douze et al. (2024). While using an IVF index reduces search time, it still requires storing the full database in memory, which for the $\approx 62$ million UniRef50 sequences requires $> 110$ GB of memory, given the 480 dimensional mean-pooled ESM-2 35M embeddings. To overcome this major challenge, further quantization is required. We rely on a *product quantizer* (PQ) to effectively reduce the dimensionality of each vector (Jégou et al., 2011). The product quantizer partitions each vector into $M$ sub-vectors, where each sub-vector is further separately quantized using a k-means clustering. Defining the product quantizer requires setting two parameters: the *code size*, $M$, and the number of bits with which to represent each sub-vector, where either 8 or 10 are commonly used.

Using a product quantizer dramatically reduces the index size. The memory requirement of indexing UniRef50 using three different code sizes and using a flat IVF index can be seen in Figure A.1. `IVFPQ32x8` refers to product-quantized IVF index, where each vector is divided into $M = 32$ sub-vectors, each of which is represented by 8 bits. The shown memory uses are *solely* for storing the index in memory. Using a quantizer such as PQ is therefore necessary in order to additionally store and train the Protriever model.

The process of preparing the coarse and product quantizers, e.g., by running k-means algorithms to facilitate fast search is called *training the index*.

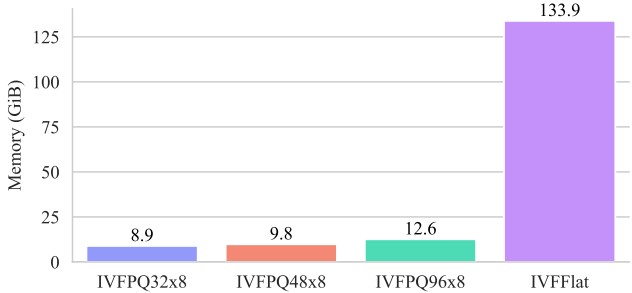

Figure A.1. Index memory use. `IVFPQ32x8` refers to product-quantized IVF index, where each vector is divided into $M = 32$ sub-vectors, each of which is represented by 8 bits, while `IVFFlat` refers to an IVF index with no further quantization. Using a product quantizer dramatically reduces the memory use, potentially at the cost of search quality.

### A.1. Distributed index

We use Protriever in a distributed setting using $N_{\text{GPU}}$ GPUs via the implementation in Izacard et al. (2022). We shard our dataset into $N_{\text{GPU}}$ equal-sized partitions and train separate indices, where each index is responsible for $N_{\text{index}} = N/N_{\text{GPU}}$ sequences. At search time, the query is used to search each index, the results of which are aggregated. We let $N_{\text{GPU}} = 4$.

### A.2. Choosing index parameters

We need to select a number of index parameters, namely the number of centroids for the coarse quantizer, $K_{\text{IVF}}$, the number of probes, $P_{\text{IVF}}$, the code size $M$, and the number of bits for the product quantizer. We have three main considerations: memory, speed, and accuracy. Given the memory uses shown Figure A.1, we now focus on gauging accuracy and speed.

#### A.2.1. RECALL

We measure search accuracy using recall by randomly sampling $N_{\text{sample}} = 10000$ UniRef50 sequences as queries and investigating whether the query sequences are returned when searching the index. We investigate code sizes of 32, 48, and 96 (the embedding dimension

needs to be divisible by the code size), as coarser quantization led to poor performance. We experiment with three different centroid counts, determined by database size: $K_{\text{IVF}} \in \{\sqrt{N_{\text{index}}}, 4\sqrt{N_{\text{index}}}, 8\sqrt{N_{\text{index}}}\}$. We fix the number of probes to $P_{\text{IVF}} = 2048$, which is the upper limit in the Faiss GPU implementation and for simplicity, we fix the number of bits per sub-vector to 8. This leads to three indices IVFPQ32x8, IVFPQ48x8, and IVFPQ96x8, with $K_{\text{IVF}} = \{3941, 15764, 31528\}$ (as $N_{\text{index}} \approx 15.5$ million). We use the 10,000 sampled queries to search across the nine index configurations, retrieving the $K = 2048$ nearest neighbors and calculating the average recall rate at powers of 2. The results can be seen in Figure A.2.

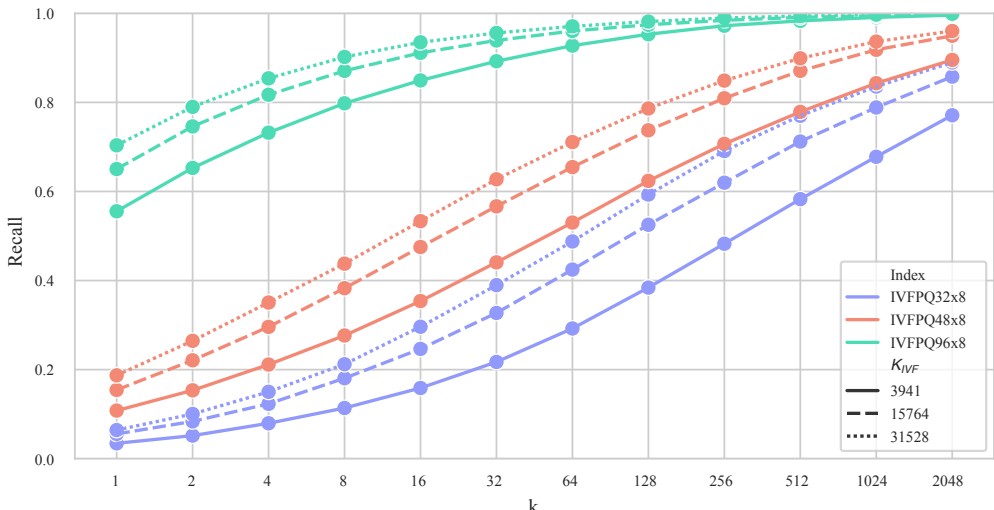

*Figure A.2.* Recall rate vs. neighborhood sizes for IVFPQ indices at different quantization levels and centroids counts. 10,000 UniRef50 sequences are randomly sampled and used as queries. For each query sequence, the 2048 nearest neighbors are found. The recall indicates whether the query sequence was successfully recovered. Decreasing the quantization from 48 sub-vectors to 96 sub-vectors leads to a significant increase in recall, while doubling the number of centroids per index from $K_{\text{IVF}} = 15764$ to $K_{\text{IVF}} = 31528$ only has a marginal performance increase.

The code size has a significant impact on search quality, where $M = 32$ (shown in blue) fails to reach recall rates above 0.9 for $K = 2048$. Increasing the code size to $M = 96$ significantly increases the recall rates, where the majority of the single-nearest neighbors return the query sequence. The search performance is less sensitive to the number of centroids, where recall increases with centroid count. For $M = 96$, we observe a persistent performance gap when using $K_{\text{IVF}} = 4\sqrt{N_{\text{index}}} = 15764$, particularly for the lower neighbor counts.

### A.2.2. SEARCH SPEED

We investigate the impact of parameter settings on search speed by measuring it over a range of scenarios. For each of the nine parameter configurations, we search using 1, 10, and 100 queries, repeating each five times. We can then visualize the search time (averaged over repeats), the time per query, and the queries per second (QPS) for each configuration. These results can be seen in Figure A.3. We see that, as expected, the search time increases with the number of queries. Using a low number of centroids (shown in blue) consistently leads to longer search times. While the search process has fewer comparisons to centroids initially, this is not outweighed by the correspondingly larger clusters. In the second row of Figure A.3 we observed that the search time per query is longer when using only a single query. This is expected as Faiss is optimized for batched searches. We also observe that using code sizes 48 and 96 approximately takes the same search time per input query. In the last row we observe that the queries per second (QPS) generally increases with the number of queries and for code size 96 appears to near a saturation point.

### A.2.3. INDEX TRAINING TIME

We lastly examine how long it takes to train the index with the different parameter configurations. We train each of the nine configurations on $N_{\text{index}} \approx 15.5$ million UniRef50 sequences a total of three times. The average training time in seconds and standard error can be seen in Table A.1. The training time is not sensitive to the code size but appears to linearly scale with number of centroids.

### A.3. Retrieval time comparison with MMseqs2

We here provide the benchmarking procedure used to generate the results in Figure 2. The tabulated results (mean and standard deviation) is additionally shown in Table A.2. We define query sizes of 1, 10, 100, and 1000 and sample a total of five query sets from UniRef50 at random for each size, resulting in 20 distinct query sets. These are then used as queries for Protriever (i.e., vector similarity search

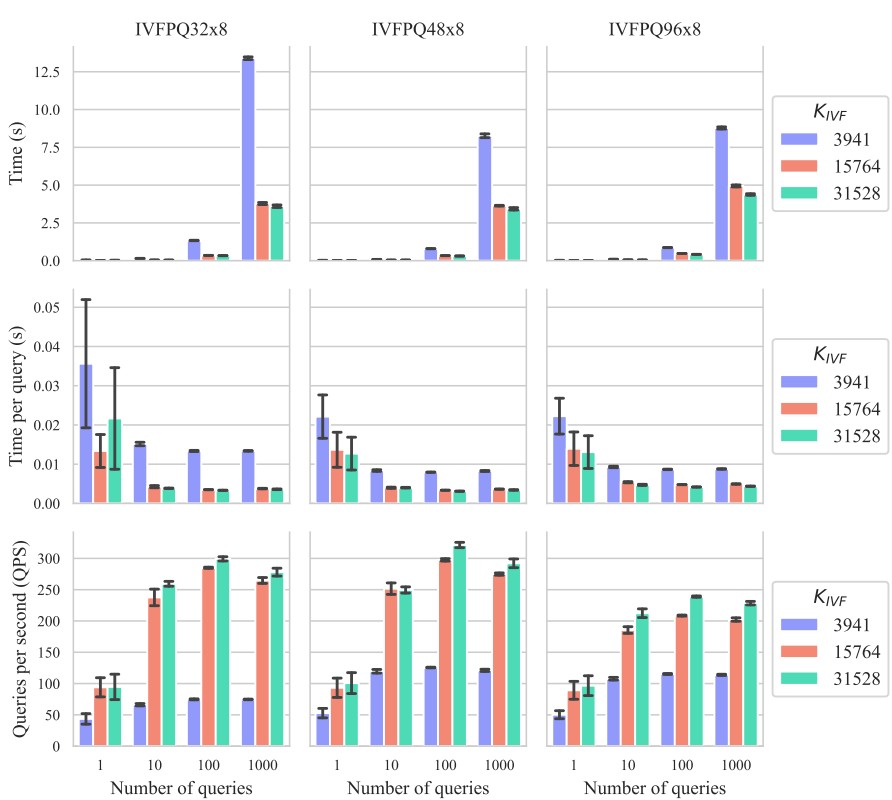

*Figure A.3.* Impact of parameter settings on search time metrics (total time, time per query, and queries per second) across 9 configurations with varying number of queries. Results show longer search times with fewer centroids, higher per-query costs for single queries due to batch optimization, and QPS improvements that scale with number of queries..

| | Training time (s) | | |
|---|---|---|---|
| | $K_{\mathrm{IVF}} = 3941$ | $K_{\mathrm{IVF}} = 15764$ | $K_{\mathrm{IVF}} = 31528$ |
| $M = 32$ | $39.00 \pm 1.36$ | $80.63 \pm 0.47$ | $190.13 \pm 0.38$ |
| $M = 48$ | $40.33 \pm 0.52$ | $82.89 \pm 0.73$ | $191.72 \pm 1.59$ |
| $M = 96$ | $47.03 \pm 0.60$ | $89.98 \pm 0.84$ | $200.13 \pm 1.69$ |

*Table A.1.* **Index training times**. Average index training times (and standard error) for different parameter configurations. Each index covers $N_{\mathrm{index}} \approx 15.5$ million UniRef50 sequences. The indexing time only slightly decreases with increased quantization. The number of centroids has a large impact on indexing time which appears to scale linearly.

through Faiss) and MMseqs2 in both CPU and GPU modes. Each search is repeated five times to account for runtime variability. To process the results, we take the median over repeated runs and compute the mean and standard deviation over the distinct query sets.

We use the following search parameters for MMseqs as described in Kallenborn et al. (2025) and detailed in the user guide:

```
mmseqs search queryDB targetDB resultDB tmp -s 8.5 -e 10000 --max-seqs 4000
--prefilter-mode 1 --threads 20
```

for the CPU-version and

```
mmseqs search queryDB targetDB_gpu resultDB tmp --gpu 1 --gpu-server 1 --db-load-mode 2
--threads 1 -e 10000 --max-seqs 4000
```

for the GPU-version.

Both query and database FASTAs are converted into MMseqs2 databases prior to searching. We additionally pre-compute the target database indices. For MMseqs2-GPU, we follow the guidelines and spawn a gpuserver via `mmseqs gpuserver targetDB_gpu --max-seqs 4000` prior to searching for reduced overhead. For Protriever, the query sequences are embedded prior to searching. The reported results thus show the time it takes for the search itself, assuming that appropriate pre-processing has been conducted for all three methods.

Protriever and GPU-accelerated MMseqs2 searches are made on a single L40S GPU using one CPU thread. The CPU-version of MMseqs2 uses 20 threads due to compute constraints. The Protriever Faiss index is pre-trained on all UniRef50 sequences and does not use sharding for direct single-GPU comparison.

| | Retrieval time per query (s) | | | |
|---|---|---|---|---|
| | $N = 1$ | $N = 10$ | $N = 100$ | $N = 1000$ |
| Protriever (Faiss) | $0.007 \pm 0.0005$ | $0.005 \pm 0.0001$ | $0.005 \pm 0.0001$ | $0.005 \pm 0.0001$ |
| MMseqs2-GPU | $0.637 \pm 0.3079$ | $0.532 \pm 0.1285$ | $0.493 \pm 0.0653$ | $0.508 \pm 0.0246$ |
| MMseqs2 (k-mer) | $28.587 \pm 5.7237$ | $17.771 \pm 2.9081$ | $15.010 \pm 1.4492$ | $15.567 \pm 1.0836$ |

*Table A.2.* **Retrieval time per query.** Average retrieval time (and standard deviation) per query as function of number of queries over various query sets.

## A.4. Retrieval homology

To investigate the homology of the retrieved sequences, we compare Protriever to MMseqs2 (Steinegger & Söding, 2017; Kallenborn et al., 2025) and JackHMMER (Potter et al., 2018). We use the ten validation sequences in ProteinGym (Notin et al., 2023) as queries for the retrieval methods. For Protriever and MMseqs2, we use the same settings as detailed in Appendix A.3, where we for JackHMMER, we use bit scores of 0.1, 0.5, and 0.9. We compute the sequence identity between query sequences and all retrieved sequences and visualize their distribution in Figure A.4. As JackHMMER returns a highly variable number of sequences, e.g., a single UniRef50 sequence was returned using `SPIKE_SARS2` as query with bit score 0.9, while 9,839 sequences were returned for `CALM1_HUMAN` with bit score 0.1, we weigh the distribution by the number of retrieved sequences for each method/query pair. This way each of the 10 retrieval sets contribute equally to the distribution visualization. During inference, retrieved sequences with less than 15 % sequence identity are removed. The resulting distributions post-filtering are shown in Figure A.5, where we now observe a gap between MMseqs2 and Protriever, where the latter retrieves sequences that are more similar to the queries.

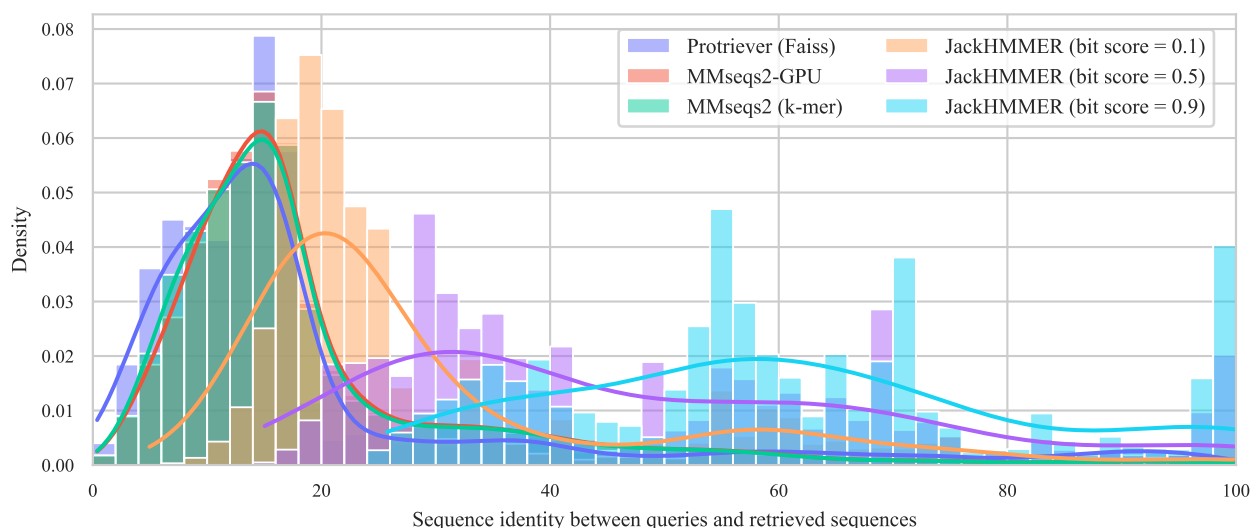

*Figure A.4.* **Distribution of sequence identities.** Distribution of sequence identities between queries and retrieved sequences for Protriever, MMSeqs2, and JackHMMER, using the ten ProteinGym validation sets as queries. The retrieved sequences have not been filtered and are weighed by number of retrieved sequences per method/query pair.

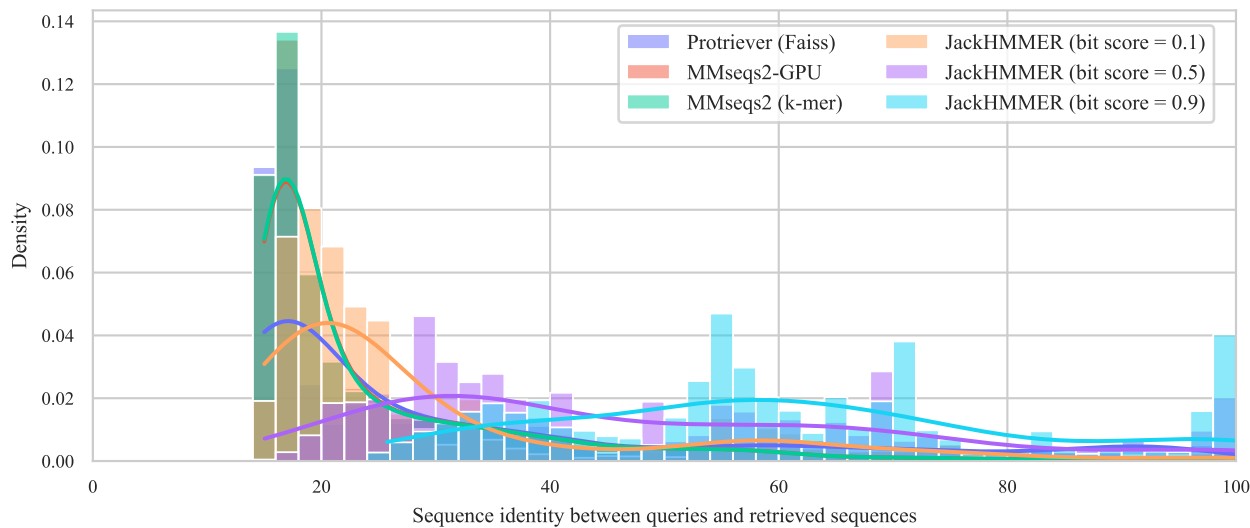

*Figure A.5.* **Distribution of sequence identities after filtering.** Distribution of sequence identities between queries and retrieved sequences for Protriever, MMSeqs2, and JackHMMER, using the ten ProteinGym validation sets as queries. The retrieved sequences have been filtered by removing those with $< 15\%$ sequence identity to the query. The retrieved sequences are weighed by number of retrieved sequences per method/query pair.

# B. Alternative architecture for the reader

This section presents results using Fusion-in-Decoder (FiD) as an alternative reader architecture, demonstrating the versatility of our retrieval framework across different architectural choices.

## B.1. Fusion-in-Decoder architecture

The Fusion-in-Decoder (FiD) model (Izacard & Grave, 2021) integrates multiple retrieved sequences by independently encoding each one and fusing their representations during decoding. Unlike PoET's approach of processing concatenated sequences, FiD encodes each retrieved sequence separately through an encoder, then concatenates their hidden representations for the decoder to attend over.

Formally, given $N$ retrieved sequences per query, each sequence is encoded independently to produce hidden states that are concatenated along the sequence dimension. During decoding, cross-attention allows the decoder to attend over all retrieved sequences simultaneously, effectively fusing information from the entire retrieved set.

Regarding computational complexity, FiD offers advantages over approaches that process all sequences jointly. While PoET processes all sequences simultaneously with quadratic complexity $O(((k+1)l)^2)$ for $k$ sequences of length $l$, FiD's complexity scales as $O(kl^2 + l^2k)$ where sequences are processed independently before cross-attention computation. This linear scaling with the number of sequences makes FiD more computationally efficient, though potentially less expressive than architectures that model interactions between all positions across all sequences simultaneously.

We implement FiD using an ESM encoder (35M parameters) and Tranception decoder (85M parameters), connected through cross-attention layers, resulting in a 150M parameter model. This approach is agnostic to the particular choice of encoder and decoder architectures, requiring only dimension alignment between encoder and decoder through projection layers.

We pretrain this FiD model on the same dataset used for PoET training, following identical sampling and data augmentation strategies. We train our model (without DPR pretraining) within the Proriever framework, with EMDR end to end loss on the retriever for 50,000 iterations. We use AdamW with a batch size of 16, a context size set to 20, learning rates of $4 \times 10^{-5}$ for the reader and $5 \times 10^{-5}$ for the retriever, with linear decay and 1,000 warm-up steps. We re-index our dataset every 5,000 steps for a total of 10 re-indexing stages.

## B.2. Fitness prediction performance of Fusion-in-Decoder architecture

We evaluate first the model with same MMseqs2 MSAs generated from uniref100 used in PoET, and using the same sampling scheme, based on (Hopf et al., 2017)(FiD + MSA in Table B.1) We also evaluate this trained MSA with a Frozen ESM retriever, which substantially degrades performance, on average going from 0.42 to 0.342. However, we are able to recapitulate the performance of the MSA input model with the trained Protriever retriever (0.416). We are in particular even better than the pretrained model evaluated with MSAs at low depth. The results in Table B.1 demonstrate that retrieval-augmented models can significantly outperform single-sequence models of comparable size (FiD with trained Protriever at 150M parameters achieves 0.416 vs ESM2-M's 0.388) and achieve performance comparable to models that are orders of magnitude larger (matching the 3B parameter ESM2-XL's 0.418 overall Spearman correlation), representing a 20-fold reduction in model size while maintaining competitive fitness prediction accuracy. Note that these results were preliminary. We could further improve these using the DPR pretraining, which showed synergistic effects.

*Table B.1.* **Zero-shot substitution DMS benchmark by MSA depth**. Average Spearman's rank correlation between model scores and experimental measurements by MSA depth on the ProteinGym substitution benchmark. Tranception models are without inference-time MSA retrieval. Alignment depth is defined by the ratio of the effective number of sequences $N_{\text{eff}}$ in the MSA, following (Hopf et al., 2017), by the length covered $L$ (Low: $N_{\text{eff}}/L$ <1; Medium: 1< $N_{\text{eff}}/L$ <100; High: $N_{\text{eff}}/L$ >100). The All column is the average across the three depths.

| Model type | Model name | # Params | Spearman by MSA depth | | | |
| --- | --- | --- | --- | --- | --- | --- |
| | | | Low | Medium | High | All |
| Encoders | ESM2-S | 35M | 0.239 | 0.271 | 0.453 | 0.321 |
| | ESM2-M | 150M | 0.306 | 0.358 | 0.500 | 0.388 |
| | ESM2-L | 650M | 0.335 | 0.406 | **0.517** | 0.419 |
| | ESM2-XL | 3B | 0.348 | **0.415** | 0.491 | 0.418 |
| Decoders | Tranception-S | 85M | 0.258 | 0.295 | 0.321 | 0.291 |
| | Tranception-M | 300M | 0.293 | 0.349 | 0.382 | 0.341 |
| | Tranception-L | 700M | 0.358 | 0.371 | 0.417 | 0.382 |
| FiD | FiD + MSA | 150M | 0.352 | 0.411 | 0.498 | **0.420** |
| | FiD + frozen Protriever | 150M | 0.287 | 0.354 | 0.386 | 0.342 |
| | FiD + trained Protriever | 150M | **0.365** | 0.401 | 0.483 | 0.416 |

# C. Detailed fitness performance results

## C.1. Protriever results by MSA depth

*Table C.1.* **Zero-shot performance segmented by MSA depth on the 217 substitution DMS of ProteinGym.** Alignment depth is defined by the ratio of the effective number of sequences $N_{\text{eff}}$ in the MSA, following (Hopf et al., 2017), by the length covered $L$ (Low: $N_{\text{eff}}/L <1$; Medium: $1< N_{\text{eff}}/L <100$; High: $N_{\text{eff}}/L >100$). $\rho$ designates Spearman rank correlation

| Model type | Model name | Low MSA depth | | Medium MSA depth | | High MSA depth | |
|---|---|---|---|---|---|---|---|
| | | $\rho$ | NDCG | $\rho$ | NDCG | $\rho$ | NDCG |
| Alignment-based | Site independent | 0.427 | 0.747 | 0.376 | 0.747 | 0.317 | 0.770 |
| | GEMME | 0.446 | 0.761 | 0.474 | 0.778 | 0.493 | 0.809 |
| | EVE | 0.420 | 0.757 | 0.457 | 0.783 | 0.477 | 0.821 |
| Unconditional pLM | ESM-1v | 0.316 | 0.685 | 0.409 | 0.743 | 0.495 | 0.808 |
| | ProGen2 | 0.323 | 0.727 | 0.412 | 0.775 | 0.442 | 0.808 |
| | ESM2 | 0.336 | 0.703 | 0.423 | 0.759 | 0.485 | 0.808 |
| Hybrid | MSA Transformer | 0.375 | 0.754 | 0.456 | 0.776 | 0.479 | 0.815 |
| | Tranception L | 0.421 | 0.762 | 0.443 | 0.778 | 0.471 | 0.812 |
| | TranceptEVE L | 0.436 | 0.764 | 0.472 | 0.785 | 0.490 | 0.824 |
| | PoET | **0.478** | 0.766 | 0.478 | 0.781 | 0.510 | 0.827 |
| Protriever | Protriever | 0.464 | **0.772** | **0.498** | **0.781** | **0.512** | **0.831** |

## C.2. Protriever results by Taxa

*Table C.2.* **Zero-shot performance segmented by Taxa on the 217 substitution DMS of ProteinGym benchmark.** $\rho$ designates spearman rank correlation.

| Model type | Model name | Human | | Other Eukaryote | | Prokaryote | | Virus | |
|---|---|---|---|---|---|---|---|---|---|
| | | $\rho$ | NDCG | $\rho$ | NDCG | $\rho$ | NDCG | $\rho$ | NDCG |
| Alignment-based | Site independent | 0.380 | 0.759 | 0.389 | 0.781 | 0.318 | 0.770 | 0.375 | 0.695 |
| | GEMME | 0.469 | 0.779 | 0.516 | 0.805 | 0.467 | 0.816 | 0.472 | 0.743 |
| | EVE | 0.454 | 0.784 | 0.495 | 0.810 | 0.457 | 0.827 | 0.434 | 0.742 |
| Unconditional pLM | ESM-1v | 0.458 | 0.770 | 0.464 | 0.768 | 0.413 | 0.797 | 0.294 | 0.641 |
| | ProGen2 | 0.386 | 0.772 | 0.458 | 0.791 | 0.418 | 0.822 | 0.402 | 0.718 |
| | ESM2 | 0.442 | 0.778 | 0.477 | 0.775 | 0.458 | 0.814 | 0.294 | 0.652 |
| Hybrid | MSA Transformer | 0.439 | 0.780 | 0.516 | 0.812 | 0.446 | 0.823 | 0.421 | 0.723 |
| | Tranception L | 0.455 | **0.788** | 0.497 | 0.807 | 0.414 | 0.812 | 0.438 | 0.727 |
| | TranceptEVE L | 0.473 | 0.787 | 0.513 | 0.816 | 0.455 | 0.831 | 0.461 | 0.743 |
| | PoET | **0.482** | 0.781 | 0.541 | **0.827** | 0.464 | 0.829 | 0.491 | **0.744** |
| Protriever | Protriever | 0.480 | **0.788** | 0.542 | 0.811 | **0.492** | **0.845** | 0.516 | **0.744** |

## D. Protriever training loss function

In addition to the EMDR loss used in our main experiments (Equation (3)), we evaluated two alternative loss functions for end-to-end retriever training.

**Perplexity Distillation (PDist)**    The PDist approach (Izacard et al., 2022) trains the retriever to predict how much each sequence improves the language model's perplexity when reconstructing the query sequence. We minimize the KL-divergence between the retriever's relevance scores (Equation (2)) and the posterior distribution based on language model performance:

$$p_k = \frac{\exp\left(\log p_{\text{LM}}\left(\mathbf{q} \mid \mathbf{d}_k\right)\right)}{\sum_{i=1}^{K} \exp\left(\log p_{\text{LM}}\left(\mathbf{q} \mid \mathbf{d}_i\right)\right)} \tag{4}$$

**Leave-One-Out Perplexity Distillation (LOOP)**    LOOP measures each sequence's contribution by evaluating how much the language model's reconstruction performance degrades when removing individual sequences from the retrieved set. For each retrieved sequence $\mathbf{d}_k$, the relevance score is defined as:

$$p_{\text{LOOP}}(\mathbf{d}_k \mid \mathbf{q}) = \frac{\exp\left(-\log p_{\text{LM}}\left(\mathbf{q} \mid \mathcal{D}_K \setminus \{\mathbf{d}_k\}\right)\right)}{\sum_{i=1}^{K} \exp\left(-\log p_{\text{LM}}\left(\mathbf{q} \mid \mathcal{D}_K \setminus \{\mathbf{d}_i\}\right)\right)} \tag{5}$$

While computationally more expensive than PDist and EMDR due to requiring $K$ separate forward passes, LOOP better reflects the multi-sequence conditioning used during training by evaluating the model on $(K-1)$ sequences rather than single sequences. Indeed, we observe in D.1 that LOOP performs the best with our FiD model in the Protriever framework. We however decided to pursue the use of EMDR for larger scale results with PoET due to its lower computational cost while maintaining competitive performance. Further work could look into LOOP as an alternative to EMDR for the PoET Protriever model.

*Table D.1.* **Spearman on validation set (Appendix F) for different losses with the FiD model**. We evaluate the FiD model with retrieved sets, sampled with the same scheme described in the main text. EMDR performs slightly better than the PDist loss. LOOP performs slightly better than the other two, but requires many more forward passes

| Training strategies | EMDR | PDist | LOOP |
|---|---|---|---|
| Frozen ESM | 0.347 | 0.347 | 0.347 |
| Protriever w/o DPR | 0.404 | 0.397 | 0.409 |

## E. Inference-time sampling of retrieved sequences

Our inference procedure involves several steps to ensure diverse and representative sequence sampling. We encode the query sequence and search the UniRef50 index for homologous sequences. From the returned set, we filter sequences that have less than 15% sequence similarity with the query. We then sample 2560 UniRef100 sequences from the filtered UniRef50 homologous set, with weights inversely proportional to the size of their corresponding UniRef90 clusters. These sequences are encoded by the retriever and subsequently clustered using $k$-means with $k = 50$.

Clusters are then sampled using weights $\sqrt{s} \cdot \left(1 + e^{-ad/T}\right)^{-1}$, where $s$ is the size of the k-mean cluster and $d$ is the distance to the query. Different combinations of $a$ and $T$ values give rise to five parameter combinations that represent different trade-offs between diversity and relevance in the retrieved set. For experiments in Section 5.2, clusters are taken as is in UniRef50, and are sampled with same weights $\sqrt{s} \cdot \left(1 + e^{-ad/T}\right)^{-1}$, where $d$ is taken as the alignment identity of the UniRef50 cluster representative to the query, and $s$ is the size of the UniRef50 cluster. We also vary the conditioning set size across three configurations (6,144, 12,288, and 24,576 tokens). Overall, we ensemble across 15 total configurations: 5 sequence diversity strategies × 3 conditioning set lengths, with each configuration requiring a separate forward pass through the model.

## F. Validation sets from ProteinGym

We use the same ProteinGym assays for validation as in Tranception (Notin et al., 2022) and PoET (Truong Jr & Bepler, 2023):

- BLAT_ECOLX (Jacquier et al., 2013)

- CALM1_HUMAN (Weile et al., 2017)

- CCDB_ECOLI (Tripathi et al., 2016)

- DLG4_RAT (McLaughlin Jr et al., 2012)

- PA_I34A1 (Wu et al., 2015)

- RL40A_YEAST (Roscoe et al., 2013)

- SPIKE_SARS2 (Starr et al., 2020)
- TPOR_HUMAN (Bridgford et al., 2020)
- Q2N0S5_9HIV1 (Haddox et al., 2018)
- SPG1_STRSG (Olson et al., 2014)

## G. Homologous database construction

We utilize the homologous sequence database constructed by Truong Jr & Bepler (2023) over UniRef50 version 2021/03 (Suzek et al., 2015). We thank the PoET authors for providing access to this valuable resource. The database was constructed through an all-vs-all homology search across UniRef50 using Diamond (Buchfink et al., 2015), a high-performance sequence alignment tool that is over 100× faster than BLAST. The search was performed with the following parameters:

```
diamond blastp -q uniref50.fasta -d diamond/uniref50 -f 6 --header -k 200000 --max-hsps 1
-e 0.001 -p 96 -o output.tab
```

This comprehensive search identifies putative homologs for each of the approximately 62 million sequences in UniRef50, creating homologous clusters that form the foundation for our retrieval experiments. The resulting cluster size distribution is shown in Figure G.1.

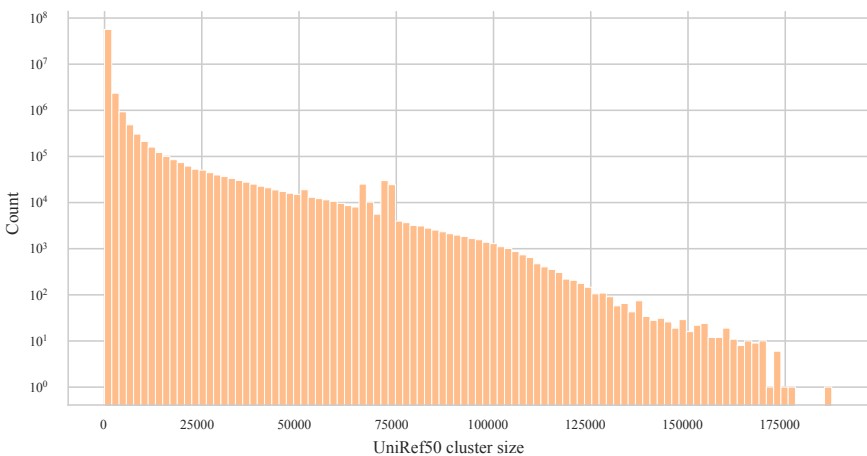

*Figure G.1.* **Distribution over cluster sizes of UniRef50.** Distribution of the $\approx 62$ million UniRef50 clusters.

## H. Software

We make our code available at https://github.com/OATML-Markslab/Protriever.

