# OpenReview forum: "Protriever: End-to-End Differentiable Protein Homology Search for Fitness Prediction"
_ICML.cc/2025/Conference — ICML 2025 poster_

### Official Review · Reviewer_cdq5 · 2025-03-12

**Overall Recommendation:** 4

**Summary:**

Protriever is an end-to-end differentiable framework for augmenting performance on downstream applications (e.g., fitness prediction)  of language models using vector-based retrieval. The model consists of 2 trainable components, the retriever and the reader, and one static vector index. The retriever learns to select from the index a set of support sequences for a query sequence while the reader learns to reconstruct the query from only the support sequences (by autoregressive decoding). In so doing, the reader learns a joint distribution over query and support sequences. After training, protriever is applied to a fitness prediction task.

**Claims And Evidence:**

The claims/result on the architectural choices are well-justified and convincing. The claim that the training procedure improves fitness prediction is supported by Tables 1 and 2, which seem to show a significant improvement over the tested baselines in only certain settings.

**Essential References Not Discussed:**

- Alongside the reference to MSAGPT, the authors may also refer to EvoDiff [Alamdari, et al '23].
- On the subject of structure-aware embeddings models (sec 5, line 410), the authors may also refer to TM-Veec [Hamamsy, et al '24, published in Nat. Bio].

**Experimental Designs Or Analyses:**

I did check the soundness/validity of the experiments.

**Methods And Evaluation Criteria:**

The evaluation criteria is sound -- the ProteinGym dataset is a well known benchmark for fitness prediction.

**Other Comments Or Suggestions:**

- Which assays were used during the ProteinGym evaluation?

**Other Strengths And Weaknesses:**

Strengths

The paper is well-written, clear, and the architectural choices are well-motivated and justified.

Weaknesses

The article emphasizes architectural decisions but the experimental results are not convincing. The model does not improve performance significantly or across-the-board. The application to fitness prediction has a well-defined benchmark, but the results do not seem to demonstrate any marked effectiveness of the model for this task.

I am not sure what Table 3 tells us, since the Spearman values are generally close. Which loss function was used in the end and why?

**Questions For Authors:**

There are many other potential applications of this model besides fitness prediction where there doesn't seem to be a clear demonstration of improved performance. I am curious whether the authors are willing to explore any other applications which could separately test the retriever and reader's capabilities. For example, both the reader and retriever can be used for function prediction (by means of homology transfer in the latter).

**Relation To Broader Scientific Literature:**

The work relates to ongoing advancements in sequence retrieval for protein property prediction, such as ProtEx [Shaw, et al '24] and builds off seminal work on retrieval augmented generation (RAG, REALM) referenced in the text.

**Theoretical Claims:**

I would like for the authors to provide a reference or brief justification of the claim in sec 3.2 line 245 that the theoretical optimum of the EMDR loss is "a degenerate distribution that assigns all probability mass to the single sequence maximizing the language model’s likelihood of generating the correct output".

---

> ### Author Rebuttal · Authors · 2025-04-01
>
> **C1: EMDR optimum**
>
> The EMDR loss function is defined as:
>
> $$ \mathcal{L}_{EMDR} = -\log \left[  \sum_k p^{LM} ( \mathbf{q} \vert \mathbf{d}_k ) p^{RETR}  ( \mathbf{d}_k \vert \mathbf{q}) \right] $$
>
> (We are experiencing issues with longer equations in OpenReview's Markdown, where subscripts appear to break the rendering - so we have resorted to put _LM_ and _RETR_ as superscripts.)
>
> Minimizing this loss is equivalent to maximizing the weighted sum inside the logarithm. Given that $p_{RETR}$ must form a valid probability distribution (summing to 1), this becomes a constrained optimization problem.
>
> The solution to such a problem is straightforward: to maximize a weighted sum $\sum_k w_k p_k$ where weights $w_k$ are fixed and $\sum_k p_k = 1$, the optimal strategy is to assign $p_j = 1$ to the term with the highest weight $w_j$, and $p_k = 0$ to all others.
>
> In our case, the weights are the language model probabilities $p_{LM}(\mathbf{q} | \mathbf{d}_k)$, so the optimal retriever distribution will place all probability mass on the document that yields the highest language model score.
>
> This is intuitive when viewing the retriever as allocating a fixed "budget" of probability mass to maximize the weighted sum — the best strategy is always to invest the entire budget in the option with the highest return.
>
> **C2: Additional References**
>
> Thank you for the suggestion — both are relevant and will be included in the revision.
>
> **C3: Validation Set**
>
> For the validation set, we have used the same set as previously used in Tranception (Notin et al) and PoET (Truong et al). It is composed of the following assays:
> - BLAT_ECOLX (Jacquier et al., 2013)
> - CALM1_HUMAN (Weile et al., 2017)
> - CCDB_ECOLI (Tripathi et al., 2016)
> - DLG4_RAT (McLaughlin et al., 2012)
> - PA_I34A1 (Wu et al., 2015)
> - RL40A_YEAST (Roscoe et al., 2013)
> - SPIKE_SARS2 (Starr et al., 2020)
> - TPOR_HUMAN (Bridgford et al., 2020)
> - Q2N0S5_9HIV1 (Haddox et al., 2018)
> - SPG1_STRSG (Olson et al., 2014)
>
> We will include the above clarification in the revision.
>
> **C4: Performance impact and breadth of tasks supported**
>
> As noted by the two other reviewers, the core practical benefit from our suggested approach lies in the significant speedups over standard retrieval methods such as multiple sequence alignments (MSA), without degrading the fitness prediction performance. We point the reviewer to the response to C2 from reviewer 2keu and C1-C2 from reviewer Bbfm for a more detailed discussion of our contributions there. Note that MSAs have been a cornerstone of computational biology for several decades and, as such, they have been extensively optimized to support tasks like the ones we focus on in this work. Thus, the types of improvements we present in this work are both non-trivial and of significant importance in practice, where these speedups matter (C3, Bbfm)
>
> We hope this addresses the concerns of the reviewer in terms of performance improvements. As for the scope of applications our paper covers, we would like to emphasize that ProteinGym encompasses 200+ assays across a wide range of protein functions (e.g,. thermostability, binding, catalytic activity, protein expression) and that, as such, our current experimental scope is already fairly broad and diverse. As noted by the reviewer, there are other protein-related applications where homology is key (e.g., function prediction, tertiary structure prediction). While an adaptation of our proposed method to these other tasks is beyond the scope of this paper, we argue that this potentiality is yet another strength of our proposed model and believe the ideas presented here will be conducive to many subsequent works that explore the benefits of end-to-end retrieval in protein modeling.

---

> > ### Comment · Reviewer_cdq5 · 2025-04-02
> >
> > I would like to thank the authors for their rebuttal and clarifying the point on the EMDR loss.
> >
> > I would however like to note that regarding C4 paragraph 2, although the ProteinGym contains data from 200+ assays surveying a wide variety of selection types, the selected validation set in C3 is heavily biased towards organismal fitness assays (7 organismal fitness [5 releated to growth, 1 amoxicillin resistance, 1 complementation], 3 binding; this was calculated by loading the DMS_substitutions.csv and DMS_indels.csv available on the ProteinGym website, searching for the strings listed in rebuttal 2keu-C1, and counting the selection type occurrences). For that reason, it is misleading to me to claim the breadth of scope as including catalytic activity, expression, and thermostability. That said, since this validation set is a well established benchmark in the literature (Notin et al, 2022 and Truong et al, 2023) and the focus of the article is on speedups that do not impact performance, I will raise my rating to an accept.

---

> > > ### Author Response · Authors · 2025-04-09
> > >
> > > We thank the reviewer for these additional comments. To address your last point of feedback, we further expanded the time-performance tradeoff analysis to the entire ProteinGym benchmark (217 assays). We obtain the following results, confirming the prior conclusions on the benefits from the proposed differentiable retrieval approach:
> > >
> > > | Retrieval Method | Retrieval time (s) | Avg. Spearman PoET (single fwd pass) | Avg. Spearman PoET (full ensembling) |
> > > |------------------|----------|--------------------------------------|--------------------------------------|
> > > | Protriever       | 0.0046   | 0.406                                | 0.436                                |
> > > | MMseqs2          | 16.860   | 0.411                                | 0.440                                |
> > > | MMseqs2-GPU      | 0.613    | 0.400                                | 0.427                                |
> > > | JackHMMER        | 2501     | 0.412                                | 0.443                                |

---

### Official Review · Reviewer_Bbfm · 2025-03-13

**Overall Recommendation:** 4

**Summary:**

The paper proposes Protriever, an end-to-end differentiable framework for protein sequence modeling. This approach unifies two steps: protein homology retrieval and downstream modeling tasks. This is done by using a vector similarity search to retrieve homologous protein sequences. The authors train Protriever end-to-end for a protein fitness prediction task. They evaluate it on the ProteinGym benchmark for zero-shot fitness prediction. Protriever achieves performance on par with traditional MSA-based approaches. The benefit is on the speedup --- it's reported to be 30-100x faster compared to existing methods.

**Claims And Evidence:**

The primary focus of the paper is on the speedup. The authors report the performance (which is indeed on par with the other methods). I think this focus is reasonable and, therefore, I won't comment on performance differences and focus mostly on the speedup part.

Comments on the speedups:

- Because we're interested in the speedup, I'd like to ensure that the results of the speedup are not influenced by any other factors. What is the benchmarking methodology for this time? What hardware was used? What is the dataset size? Configuration parameters?

- On Figure 2, can you report more queries than 1 and 100? Why these numbers? Can you also report uncertainty and confidence intervals?

- Because the primary benefit seems to be the speedup, could you tell me and report the end-to-end performance timing for complete prediction tasks? Right now, the speedup is only for the retrieval time. I think it could be alright if the training time is much larger for your method than other methods (i.e. I think you can make the case that retrieval time is more important), but I'd like to have this analysis. So: (i) can you compare the end-to-end time?; (ii) if the end-to-end time for your application is larger with one retrieval, how many retrievals would you need to have for you to break-even time-wise?

- On a more practical level, could you tell me a bit more about what applications require such speedups (i.e. are there any reports showing that speedups are issues people are trying to solve?). I'm happy if these are non-academic reports too if that helps.

Other comments:

- On ablation studies. The ablation studies (Table 3, Table 4) seem to have very minor differences between different values tested. Could you report any uncertainty intervals (e.g. standard deviation over different splits)? It's hard to see whether the values are significant.

- The statistical significance angle is true elsewhere too. For instance, you highlight Protriever's advantage in low MSA depth regimes (0.365 vs. 0.352), but this small difference may not be statistically significant, and the paper doesn't provide statistical validation. Because you are reporting average Spearman rank correlation, I don't have good intuition whether such a difference means anything -- to me this looks like noise.

- You claim that Protriever is "architecture-agnostic," but this appears somewhat overstated as the flexibility to substitute different components is common to many deep learning frameworks. I appreciate the separation of retriever and reader components in your design,  but it's somewhat of a trivial point that these could be any architectures. The same is true for the task. I'd recommend either removing this framing or providing empirical evidence across multiple tasks and architectures beyond fitness prediction to make this more of an important point in the paper.

**Essential References Not Discussed:**

N/A.

**Experimental Designs Or Analyses:**

I looked at the experimental design. The issues I've noticed are already included above.

**Methods And Evaluation Criteria:**

The proposed methods and evaluation criteria are generally appropriate. ProteinGym is a suitable and standard choice in protein fitness prediction.

While most things are clear, here are a few questions from my end that it would be great to get answers to:

- One issue is that the paper evaluates only on single amino acid substitutions and doesn't test more complex mutation scenarios (e.g. deletions, insertions).

- Another issue -- repeating the comment from before -- is that the authors focus only focus on a single prediction task, so it limits your claim that this is task-agnostic (even though I understand what you mean by this).

- It would be also great if you could expand your writeup to explain what you do (e.g. training procedures) instead of simply stating that you use PoET's dataset and sampling approach (which makes it hard for a reader to follow what's happening).

- Also, do you have any intuition why different sampling strategies work better than others?

**Other Comments Or Suggestions:**

N/A

## Post-rebuttal update: Increased 3 -> 4 based on the reply.

**Other Strengths And Weaknesses:**

Some strengths I haven't mentioned before:

(1) the framework's ability to change retrieval databses at inference time seems to be practically useful;

(2) Generally, it seems to integrate multiple disciplines quite well. The EMDR and PDist training approaches represent interesting adaptations of NLP retrieval methods to the protein domain, with EMDR showing slightly better performance in the ablation studies. The Fusion-in-Decoder architecture is a sensible choice that handles the computational complexity challenges of processing multiple sequences effectively.

(3) it seems to address clear problems within MSA. The end-to-end differentiable retrieval framework for protein sequences represents a novel technical contribution that hasn't been previously explored in this domain.

The drawbacks are largely those that relate to the questions I had asked previously.

**Questions For Authors:**

Asked above.

**Relation To Broader Scientific Literature:**

Protriever builds upon conditional protein language models like MSA Transformer (Rao et al., 2021) and PoET (Truong Jr & Bepler, 2023), which condition on multiple related sequences. Unlike MSA Transformer, which requires pre-aligned sequences and cannot model insertions/deletions, Protriever follows PoET's approach of handling unaligned sequences, but adds the innovation of AIDO which sequences are most useful during training.

Protriever adapts RAG techniques from NLP (Lewis et al., 2020) to protein modeling, similar to concurrent work like RSA (Ma et al., 2024) and AIDO.RAG (Li et al., 2024).

**Theoretical Claims:**

No proofs or theoretical claims.

---

> ### Author Rebuttal · Authors · 2025-04-01
>
> **C1: Speed-up: inference analysis**
>
> Instead of reporting per query times from the MMseqs-GPU paper from Kallenborn et al. we move to rigorously benchmark our method, MMseqs2 in CPU and GPU modes.
> We randomly sample UniRef50 sequences to create 5 sets of sizes 1, 10, 100, and 1000 each. For each query size, we report the mean and standard deviation across the 5 random datasets (after taking the median over 5 repeated runs per dataset).
> We use the same MMseqs2 parameters as described in Kallenborn & Chacon et al. to search against ~62 million UniRef50 sequences: increased sensitivity for the CPU-version (s=8.5), prefilter-mode 1, E-value cutoff 10000, and 4000 max sequences. For the GPU-version, we use flags ‘--gpu 1’ and ‘--gpu-server 1’, and the same E-value cutoff and max number of sequences.
>
> Our method and MMseqs2-GPU are run on identical hardware with a single L40S using a single thread. MMseqs2 uses 20 threads. The Protriever faiss index is pre-trained on all sequences and does not use sharding for direct single-GPU comparison.
>
> Search time per query (in seconds) shows that Protriever remains orders of magnitude faster across query sizes
>
> | Method| N=1| N=10| N=100| N=1000|
> |:-|:-|:-|:-|:-|
> | MMseqs2 (k-mer)| 28.587 ± 5.7237 | 17.771 ± 2.9081 | 15.010 ± 1.4492 | 15.567 ± 1.0836|
> | MMseqs2-GPU| 0.637 ± 0.3079| 0.532 ± 0.1285|0.493 ± 0.0653|0.508 ± 0.0246|
> | Protriever (faiss) | 0.007 ± 0.0005| 0.005 ± 0.0001| 0.005 ± 0.0001| 0.005 ± 0.0001|
>
> **C2: Speed-up: training analysis**
>
> Embedding the full UniRef50 database with ESM-2 (35M) takes less than 2 hours on a single GPU when using efficient flash-attention implementations. A full training and inference time comparison with other retrieved methods is practically not possible, each requiring dedicated processing steps (MMseqs2 costly database indexing, traditional homology methods compute expensive MSA profiles). Faiss's advantage is that we can export a small index that represents a “cached” result, which can easily be downloaded by end users. Fair comparisons should focus on inference time, where our method demonstrates clear superiority, as shown above. In response C1 to reviewer 2keu, we have included a table showing the performance and retrieval times of competing methods on 10 validation sets from ProteinGym. The faiss-driven retrieval time is several orders of magnitude faster, such that substituting it with MMseqs2-GPU would increase the computation time significantly.
>
> **C3: Applications benefiting from retrieval speedups**
>
> There are many applications that could significantly benefit from the architecture we present in this work. To name a few:
> - Evaluating the effect of missense and indel mutants on the full human proteomes entails extracting 20.000+ MSAs, one for each gene, which is time and compute heavy.
> - Training structure prediction models such as AlphaFold or OpenFold. The Open Protein set that was used to train the latter is for instance composed of 16 million MSAs. Refreshing this set as underlying data sources keep being updated is intensive.
> - Rapid retrieval is similarly useful in the inference setting of structure prediction models that leverage homologous sequences, where the arguments made in Kallenborn et al. similarly apply here.
>
> **C4: Statistical significance**
>
> We calculate non-parametric bootstrap standard errors (BSE), methodology from ProteinGym. As results in Tables 1 and 2 are reported over large number of assays they are robustly statistically significant (low MSA depth BSE<0.001). For the the standard estimate for EMDR vs. PDist as computed over the validation set only, the statistical significance is of 0.0072.
>
> **C5: Insertion/Deletion**
>
> Thank you for the suggestions. We have evaluated our Fusion-in-Decoder on the set of 66 assays in the ProteinGym benchmark. We obtain a score of 0.425 for FiD evaluated with MSA retrieval and 0.422 for FiD with Protriever retrieval. We can observe once again that we match (BSE=0.01) performance of the standard MSA-retrieval approach further demonstrating the versatility of the proposed framework.
>
> **C6: Sampling Strategies at training and inference**
>
> During training, sequences are sampled with weight inversely proportional to Uni50 cluster sizes to avoid oversampling large clusters (will have more hits at the Uni50 level). We also sample sequences from Uni100 for each Uni50 sequence with weight inversely proportional to the size of the Uni90 clusters they belong to introduce more diversity and not overly represent larger Uni90 clusters.
> At inference we test out different sampling strategies that combine two sampling strategies:
> - When sampling Uni100 with Uni90 sequence weights, as in training we make sure to not oversample large Uni90 clusters within each Uni50 representative.
> - When sampling based on embedding distance, we aim to sample closer sequences at Uni50 level, our retriever is trained to put closest to each query the most useful retrieved documents.

---

### Official Review · Reviewer_2keu · 2025-03-13

**Overall Recommendation:** 4

**Summary:**

This paper presents a method for jointly training a homology retrieval model and a conditional language model for fitness prediction. The retrieval model efficiently identifies homologous sequences for a given query protein using efficient vector search.

**Claims And Evidence:**

The paper clearly articulates its main claim, and the empirical study aligns well with it. However, it lacks sufficient comparison with the state-of-the-art model, PoET, which is mentioned in the related work.

Additionally, while the paper asserts that the proposed method achieves strong performance with improved computational efficiency, the provided empirical studies do not convincingly support this claim. A more compelling demonstration would be a curve plotting computation time (x-axis) against performance (Spearman rank correlation, y-axis), clearly illustrating the model’s advantage in the time-performance trade-off.

**Essential References Not Discussed:**

The paper appropriately cites essential references. However, as mentioned earlier, a more in-depth discussion comparing the proposed method with the SOTA, particularly PoET, should be included to provide a clearer contextual understanding of its contributions and limitations.

**Experimental Designs Or Analyses:**

This paper lacks sufficient comparison with the baselines. For example, the SOTA methods in the ProteinGym are ProSST and PoET.  ProSST uses structure information, which makes it an unfair comparison with this paper. However, the PoET is closely related, and the performance is better than the performance of this model, which should be compared and discussed.

This paper lacks sufficient comparison with relevant baselines. In particular, the SOTA method PoET in ProteinGym is not adequately discussed. PoET is closely related and achieves superior performance. A thorough comparison with PoET, including a discussion of the differences and potential advantages, would strengthen the paper.

**Methods And Evaluation Criteria:**

The proposed method is intuitive, though the EMDR retriever loss does not appear to be an EM objective. While this may not be a major contribution of the paper, it would be helpful for the authors to provide an explanation—perhaps in the Appendix—to clarify the validity of this objective.

The evaluation criteria follow the standard setting and seem appropriate.

**Other Comments Or Suggestions:**

NA

**Other Strengths And Weaknesses:**

This paper demonstrates a fair level of originality. While the methods employed originate from prior work in other domains, such as NLP, this is the first study to apply them to protein fitness prediction, marking a novel contribution to this field.

**Questions For Authors:**

(1) Why is the proposed method not compared with PoET in Table 1?

(2) In the last paragraph of Sec. 4.3, it states: 'Each method was initialized from the checkpoint of the previous method.' Does this mean that the model using the 'ESM sets' strategy was trained based on the checkpoints from the 'Fixed BLAST sets' strategy? If so, is this a fair comparison for evaluating the incremental performance of the different training strategies?

I will consider raising my score if the authors address these two questions, particularly the first one.

**Relation To Broader Scientific Literature:**

This paper is related to scientific literature about protein fitness predictions.

**Theoretical Claims:**

No theoretical claims are made in this paper.

---

> ### Author Rebuttal · Authors · 2025-04-01
>
> **C1: PoET / SOTA results**
>
> Our suggested differentiable retriever approach can readily be used to augment SOTA sequence-based architectures on the ProteinGym benchmark such as PoET. The framework requires minimal changes, though the code has to be modified to allow each query in the batch to process a variable number of retrieved sequences.
> PoET leverages additional processing of sequences based on various sequence similarity cutoffs on the aligned sequences. This results in more than 30 forward passes through the model (sequences scored from N to C and C to N, with 15 different sampling hyperparameters). Note that PoET's performance when scoring only one direction with default parameters (context size of 12288 and sequence similarity cutoff of 0.95) on the validation set is 0.425. While we could integrate an alignment step in our framework, which would allow for similar inference results, we present here results of a single forward pass of the model, which we think better represents the true potential of the method.
> We focus our analysis on the subset of 10 DMS assays from ProteinGym used as our validation set, and obtain the following results:
>
> | Assay | Spearman - vanilla PoET | Spearman - PoET w/ Protriever |
> |-|-|-|
> | BLAT_ECOLX_Jacquier_2013 | 0.677 | 0.700 |
> | CALM1_HUMAN_Weile_2017 | 0.201 | 0.208 |
> | CCDB_ECOLI_Tripathi_2016 | 0.423 | 0.422 |
> | DLG4_RAT_McLaughlin_2012 | 0.594 | 0.581 |
> | PA_I34A1_Wu_2015 | 0.527 | 0.526 |
> | Q2N0S5_9HIV1_Haddox_2018 | 0.541 | 0.518 |
> | RL40A_YEAST_Roscoe_2013 | 0.420 | 0.477 |
> | SPG1_STRSG_Olson_2014 | 0.272 | 0.216 |
> | SPIKE_SARS2_Starr_2020_binding | 0.357 | 0.356 |
> | TPOR_HUMAN_Bridgford_2020 | 0.447 | 0.395 |
> | **Average** | **0.446** | **0.440** |
>
> We match the fitness prediction performance of PoET obtained with ColabFold (performance difference is not statistically significant; using the same bootstrap standard error (BSE) methodology from ProteinGym, BSE=0.0092)
>
> **C2: time-performance trade-off**
>
> Thank you for the great suggestion.
> We compared the timing and fitness prediction performance from 4 sequence retrieval systems (Protriever, MMseqs2 (k-mer), MMseqs2-GPU, and JackHMMER) used in conjunction with our proposed FiD architecture or PoET (as discussed in the prior response). For the MMseqs2 searches, we use the default parameters from Kallenborn & Chacon et al. (see our response on benchmarking methodology B1 to reviewer Bbfm for details).
> We query each wild type sequence of the validation set against the same UniRef50 database as the one used in the original PoET paper (i.e., the 2023_04 checkpoint)
> We obtain the following results on the validation set, using either the FiD model or PoET (single inference forward pass) :
>
> | Retrieval Method | Avg. Retrieval Speed (sec) | Avg. Spearman FiD | Avg. Spearman PoET |
> |-|-|-|-|
> | Protriever| 0.0046  |0.404| 0.440
> | MMseqs2| 25.9495| 0.411|0.446
> | MMseqs2-GPU| 0.7745| 0.402| 0.435
> | JackHMMER| 2645.0| 0.413| 0.449
> We observe that Protriever achieves more than 100x speedups in retrieval, while maintaining comparable fitness prediction performance.
>
> **C3: EMDR as EM like objective**
>
> Our EMDR loss follows Sachan et al.'s "EM-inspired" (we use the same terminology as the authors) approach rather than classical EM. The connection to EM is:
>
> 1) Implicit E-step: Latent variables are the relevant documents. We estimate their posterior distribution by computing $p_{RETR}(\mathbf{d}_k|\mathbf{q})$ for retrieved documents.
>
> 2) Implicit M-step: We update parameters by maximizing the loss in our paper.
>
> The loss is akin to EM's core concept of alternating between estimating latent document relevance and optimizing model parameters.
>
> **C4: Training strategies**
>
> The ESM set strategy was initialized from the last fixed BLAST experiment checkpoint. These results are not set for comparison but to show how these three strategies of iterative training allow us to improve the performance of our retriever. Fixed BLAST sets provide the biggest boost in performance compared to the base retriever (0.347 to 0.379 for EMDR), and allows for the retriever to focus training on hard-to-distinguish sequences, as these are pulled by sequence similarity BLAST search. The second round of training with ESM sets contains easier to separate sequences and further increases performance (0.379 to 0.385). Finally, we train the reader too, end to end on the learned retriever sets, so that the reader model learns to extend to further sequences (0.385 to 0.404).

---

> > ### Comment · Reviewer_2keu · 2025-04-02
> >
> > I don’t think it’s fair to compare your approach with a degenerated version of PoET. A more compelling evaluation would be to integrate your method with the full version of PoET.
> >
> > However, the time-performance trade-off clearly highlights your contributions. Given this, I will raise my score.

---

> > > ### Author Response · Authors · 2025-04-09
> > >
> > > Thank you for the additional comments! To your last point regarding the comparison with PoET, we further expanded the analysis to strictly match all inference optimization details from the original PoET paper, namely:
> > > - Sampling of sequences based on sequence similarity weights as described in Hopf et al [1];
> > > - Ensembling across all possible combinations of 5 different max similarity cutoffs to the query (1.0, 0.95, 0.90, 0.70, 0.50) and 3 maximum context lengths (6144, 12288, 24576), i.e., 15 combinations total;
> > > - For each of these combinations, scoring sequences from N to C and from C to N, and averaging the corresponding scores.
> > >
> > > We report below an updated performance comparison on the same validation set as before:
> > > | Retrieval Method | Avg. Spearman PoET (single fwd pass) | Avg. Spearman PoET (full ensembling) |
> > > |------------------|--------------------------------------|--------------------------------------|
> > > | Protriever PoET  | 0.440                                | 0.468                                |
> > > | Vanilla PoET     | 0.446                                | 0.470                                |
> > >
> > > In that setting, we also find that Protriever PoET matches the fitness prediction performance of vanilla PoET (bootstrap standard error = 0.0071), while being several orders of magnitude faster to retrieve the set of homologous sequences as discussed before.
> > >
> > > Please let us know if you have any outstanding comments. We sincerely thank you for your valuable input on the manuscript.
> > >
> > > [1] Hopf, Thomas A. et al. “Mutation effects predicted from sequence co-variation.” Nature Biotechnology 35 (2017)

---

### Decision · Program_Chairs · 2025-05-01

**Decision:**

Accept (poster)

**Comment:**

The paper introduces Protriever, an end-to-end differentiable framework for protein sequence modeling that unifies protein homology retrieval and fitness prediction. Protriever employs a vector similarity search to efficiently retrieve homologous sequences and trains the retriever and reader components jointly for fitness prediction. The framework achieves performance comparable to traditional MSA-based methods while being 30-100x faster, as demonstrated on the ProteinGym benchmark for zero-shot fitness prediction. Although heavily borrowing from prior work in NLP, this is the first work to apply joint retrieval and conditional language modeling to protein fitness prediction, which is significantly faster than traditional MSA-based approaches.

All reviewers unanimously agree to accept the paper after the rebuttal phase, so is the final decision.